# Social-affective features drive human representations of observed actions

**Diana C Dima[1]\*, Tyler M Tomita[2], Christopher J Honey[2], Leyla Isik[1]**

[1]Department of Cognitive Science, Johns Hopkins University, Baltimore, United States; [2]Department of Psychological and Brain Sciences, Johns Hopkins University, Baltimore, United States

**Abstract** Humans observe actions performed by others in many different visual and social settings. What features do we extract and attend when we view such complex scenes, and how are they processed in the brain? To answer these questions, we curated two large-scale sets of naturalistic videos of everyday actions and estimated their perceived similarity in two behavioral experiments. We normed and quantified a large range of visual, action-related, and social-affective features across the stimulus sets. Using a cross-validated variance partitioning analysis, we found that social-affective features predicted similarity judgments better than, and independently of, visual and action features in both behavioral experiments. Next, we conducted an electroencephalography experiment, which revealed a sustained correlation between neural responses to videos and their behavioral similarity. Visual, action, and social-affective features predicted neural patterns at early, intermediate, and late stages, respectively, during this behaviorally relevant time window. Together, these findings show that social-affective features are important for perceiving naturalistic actions and are extracted at the final stage of a temporal gradient in the brain.

## Editor's evaluation

This study investigates and characterizes the representations of visual actions in video stimuli. The combination of the analytical techniques and stimulus domain makes the article likely to be of broad interest to scientists interested in action representation amidst complex sequences. This article enhances our understanding of visual action representation and the extraction of such information in natural settings.

\*For correspondence: ddima@jhu.edu

## Introduction

In daily life, we rely on our ability to recognize a range of actions performed by others in a variety of different contexts. Our perception of others' actions is both efficient and flexible, enabling us to rapidly understand new actions no matter where they occur or who is performing them. This understanding plays a part in complex social computations about the mental states and intentions of others (*Jamali et al., 2021*; *Spunt et al., 2011*; *Thornton et al., 2019*; *Thornton and Tamir, 2021*; *Weaverdyck et al., 2021*). Visual action recognition also interacts cross-modally with language-based action understanding (*Bedny and Caramazza, 2011*; *Humphreys et al., 2013*). However, there are two important gaps in our understanding of action perception in realistic settings. First, we still do not know which features of the visual world underlie our representations of observed actions. Second, we do not know how different types of action-relevant features, ranging from visual to social, are processed in the brain, and especially how they unfold over time. Answering these questions can shed light on the computational mechanisms that support action perception. For example, are different semantic and social features extracted in parallel or sequentially?

Relatively few studies have investigated the temporal dynamics of neural responses to actions. During action observation, a distributed network of brain areas extracts action-related features ranging from visual to abstract, with viewpoint-invariant responses emerging as early as 200 ms (*Isik et al., 2018*). Visual features include the spatial scale of an action (i.e., fine-scale manipulations like knitting vs. full-body movements like running) represented throughout visual cortex (*Tarhan and Konkle, 2020*), and information about biological motion, thought to be extracted within 200 ms in superior temporal cortex (*Giese and Poggio, 2003*; *Hirai et al., 2003*; *Hirai and Hiraki, 2006*; *Johansson, 1973*; *Jokisch et al., 2005*; *Vangeneugden et al., 2014*). Responses in occipito-temporal areas have been shown to reflect semantic features like invariant action category (*Hafri et al., 2017*; *Lingnau and Downing, 2015*; *Tucciarelli et al., 2019*; *Tucciarelli et al., 2015*; *Wurm and Caramazza, 2019*; *Wurm and Lingnau, 2015*), as well as social features like the number of agents and sociality of actions (*Tarhan and Konkle, 2020*; *Wurm et al., 2017*; *Wurm and Caramazza, 2019*).

Among the visual, semantic, and social features thought to be processed during action observation, it is unclear which underlie our everyday perception in naturalistic settings. Mounting evidence suggests that naturalistic datasets are key to improving ecological validity and reliability in visual and social neuroscience (*Haxby et al., 2020*; *Nastase et al., 2020*; *Redcay and Moraczewski, 2020*). Most action recognition studies to date have used controlled images and videos showing actions in simple contexts (*Isik et al., 2018*; *Wurm and Caramazza, 2019*). However, presenting actions in natural contexts is critical as stimulus–context interactions have been shown to modulate neural activity (*Willems and Peelen, 2021*). Recent attempts to understand naturalistic action perception, however, have yielded mixed results, particularly with regard to the role of social features. For example, one recent study concluded that sociality (i.e., presence of a social interaction) was the primary organizing dimension of action representations in the human brain (*Tarhan and Konkle, 2020*). Another, however, found that semantic action category explained the most variance in fMRI data, with little contribution from social features (*Tucciarelli et al., 2019*).

Here, we combined a new large-scale dataset of everyday actions with a priori feature labels to comprehensively sample the hypothesis space defined by previous work. This is essential in light of the conflicting results from previous studies, as it allowed us to disentangle the contributions of distinct but correlated feature spaces. We used three-second videos of everyday actions from the "Moments in Time" dataset (*Monfort et al., 2020*) and replicated our results across two different stimulus sets. Action videos were sampled from different categories based on the American Time Use Survey (*ATUS, 2019*) and were highly diverse, depicting a variety of contexts and people. We quantified a wide range of visual, action-related, and social-affective features in the videos and, through careful curation, ensured that they were minimally confounded across our dataset.

We used this dataset to probe the behavioral and neural representational space of human action perception. To understand the features that support natural action viewing, we predicted behavioral similarity judgments using the visual, action-related, and social-affective feature sets. Next, to investigate the neural dynamics of action perception, we recorded electroencephalography (EEG) data while participants viewed the stimuli, and we used the three sets of features to predict time-resolved neural patterns.

We found that social-affective features predict action similarity judgments better than, and independently of, visual and action-related features. Visual and action-related features explained less variance in behavior, even though they included fundamental features such as the scene setting and the semantic category of each action. Neural patterns revealed that behaviorally relevant features are automatically extracted by the brain in a progression from visual to action to social-affective features. Together, our results reveal the importance of social-affective features in how we represent other people's actions, and show that these representations emerge in the brain along a temporal gradient.

## Results
### Disentangling visual, action, and social-affective features in natural videos

We curated two sets of naturalistic three-second videos of everyday actions from the Moments in Time dataset (*Monfort et al., 2020*). The videos were selected from a larger set, ensuring that features of interest were minimally correlated. 18 common activities based on the National Bureau of Labor

**Table 1.** Activities from the American Time Use Survey (ATUS) included in each of the two stimulus sets, with the amount of daily hours spent performing each activity and the corresponding verb labels from the Moments in Time dataset.

Note that control videos were only included in the first dataset. Fighting and hiking were added for variation in valence and action setting.

| Activity | Hours | Verb labels (Moments in Time) |
|---|---|---|
| Childcare/taking care of children | 0.37 | Crying, cuddling, feeding, giggling, socializing |
| Driving | 1.17 | Driving, socializing |
| Eating | 1.06 | Chewing, eating |
| Fighting | | Fighting |
| Gardening | 0.17 | Gardening, mowing, planting, shoveling, weeding |
| Grooming | 0.68 | Bathing, brushing, combing, trimming, washing |
| Hiking | | Hiking |
| Housework | 0.53 | Cleaning, dusting, repairing, scrubbing, vacuuming |
| Instructing and attending class | 0.22 | Instructing, teaching |
| Playing games | 0.26 | Gambling, playing+fun, playing+videogames, socializing |
| Preparing food | 0.60 | Barbecuing, boiling, chopping, cooking, frying, grilling, rinsing, stirring |
| Reading | 0.46 | Reading |
| Religious activities | 0.14 | Praying, preaching |
| Sleeping | 8.84 | Resting, sleeping |
| Socializing and social events | 0.64 | Celebrating, dancing, marrying, singing, socializing, talking |
| Sports | 0.34 | Exercising, playing+sports, swimming, throwing |
| Telephoning | 0.16 | Calling, telephoning |
| Working | 3.26 | Working |
| Control videos | | Blowing, floating, raining, shaking |

Statistics' American Time Use Survey (**ATUS, 2019**) were represented (**Table 1**; see section 'Behavior: Stimuli'). The two stimulus sets contained 152 videos (eight videos per activity and eight additional videos with no agents, included to add variation in the dataset, see section 'Behavior: Stimuli') and 65 videos (three or four videos per activity), respectively. The second set was used to replicate behavioral results in a separate experiment with different stimuli and participants.

Naturalistic videos of actions can vary along numerous axes, including visual features (e.g., the setting in which the action takes place or objects in the scene), action-specific features (e.g., semantic action category), and social-affective features (e.g., the number of agents involved or perceived arousal). For example, an action like 'eating' may vary in terms of context (in the kitchen vs. at a park), object (eating an apple vs. a sandwich), and number of agents (eating alone vs. together). Drawing these distinctions is crucial to disambiguate between context, actions, and agents in natural events. To evaluate these different axes, we quantified 17 visual, action-related, and social-affective features using image properties, labels assigned by experimenters, and behavioral ratings collected in online experiments (**Figure 1a**). Visual features ranged from low-level (e.g., pixel values) to high-level features related to scenes and objects (e.g., activations from the final layer of a pretrained neural network). Action-related features included transitivity (object-relatedness), activity (the amount of activity in a video), effectors (body parts involved), and action category based on the ATUS (**ATUS, 2019**). Finally, social-affective features included sociality, valence, arousal, and number of agents (see section 'Representational similarity analysis'). Representational dissimilarity matrices (RDMs) were created for each feature by calculating pairwise Euclidean distances between all videos.

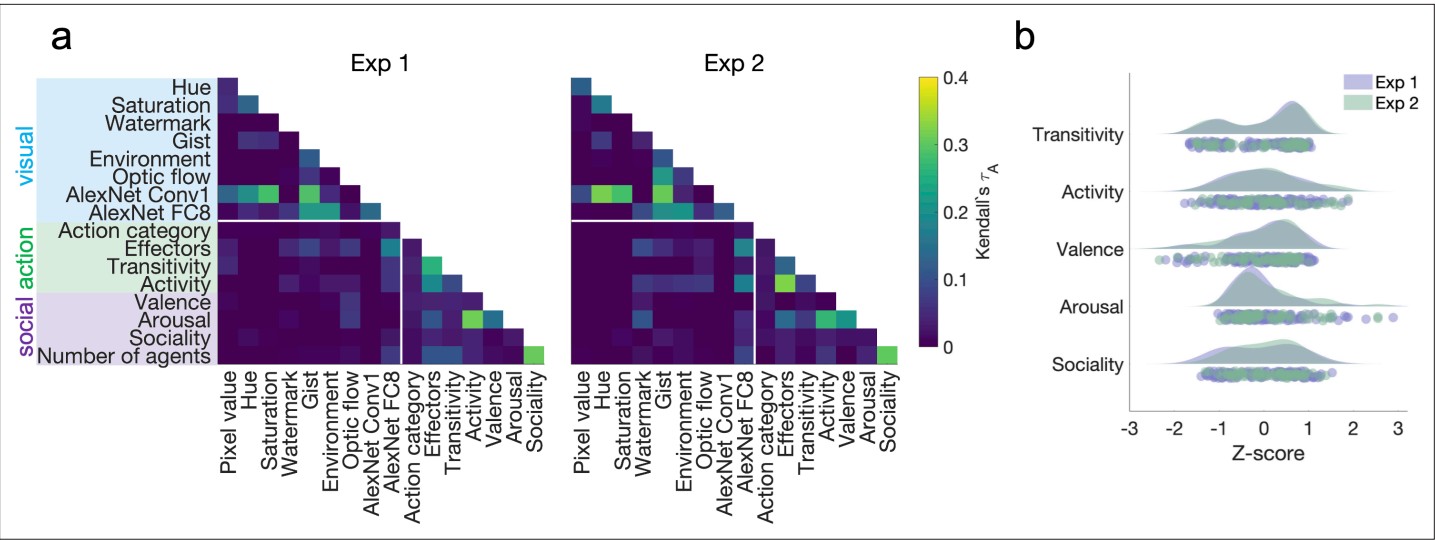

**Figure 1.** Quantifying visual, social-affective, and action features in the two stimulus sets. (**a**) Correlations between feature representational dissimilarity matrices (RDMs). Note the low correlations between visual features and action/social-affective features (white rectangle). (**b**) Behavioral rating distributions in the two stimulus sets. The z-scored ratings were visualized as raincloud plots showing the individual data points, as well as probability density estimates computed using MATLAB's *ksdensity* function (*Allen et al., 2019*).

In both video sets, there were only weak correlations between visual features and the higher-level action/social-affective features (*Figure 1a*). The highest correlations were those within each of the three sets of features, including visual features (Experiment 1: Conv1 and image saturation/gist, $\tau_A$ = 0.29; Experiment 2: Conv1 and image hue, $\tau_A$ = 0.32), action features (Experiment 1: arousal and activity, $\tau_A$ = 0.31; Experiment 2: activity and effectors, $\tau_A$ = 0.33), and social features (sociality and number of agents; Experiment 1: $\tau_A$ = 0.31, Experiment 2: $\tau_A$ = 0.3).

The distributions of action and social-affective features were not significantly different between the two stimulus sets (all Mann–Whitney $z$ < 1.08, p>0.28). The width of these distributions suggests that the stimuli spanned a wide range along each feature (*Figure 1b*). In both experiments, transitivity was notable through its bimodal distribution, likely reflecting the presence or absence of objects in scenes, while other features had largely unimodal distributions.

Behaviorally rated features differed in reliability in Experiment 1 ($F(4,819)$ = 22.35, p<0.001), with sociality being the most reliable and arousal the least reliable (*Figure 3—figure supplement 1*). In Experiment 2, however, there was no difference in reliability ($F(4,619)$ = 0.76, p=0.55). Differences in reliability were mitigated by our use of feature averages to generate feature RDMs.

## Individual feature contributions to behavioral similarity

To characterize human action representations, we measured behavioral similarity for all pairs of videos in each set in two multiple arrangement experiments (see section 'Multiple arrangement'). Participants arranged videos according to their similarity inside a circular arena (*Figure 2*). The task involved arranging different subsets of 3–8 videos until sufficiently reliable distance estimates were reached for all pairs of videos. Videos would play on hover, and participants had to play and move each video to proceed to the next trial. In Experiment 1, participants arranged different subsets of 30 videos out of the total 152, while in Experiment 2, participants arranged all 65 videos. To emphasize natural behavior, participants were not given specific criteria to use when judging similarity. Behavioral RDMs containing the Euclidean distances between all pairs of stimuli were reconstructed from each participant's multiple arrangement data using inverse MDS (*Kriegeskorte and Mur, 2012*).

The multiple arrangement task was unconstrained, which meant that participants could use different criteria. Although this may have introduced some variability, the adaptive algorithm used in the multiple arrangement task enabled us to capture a multidimensional representation of how actions are intuitively organized in the mind, while at the same time ensuring sufficient data quality. Data reliability was quantified using leave-one-subject-out correlations of the dissimilarity estimates and was above chance in both experiments (Kendall's $\tau_A$ = 0.13 ± 0.08 and 0.18 ± 0.08 respectively,

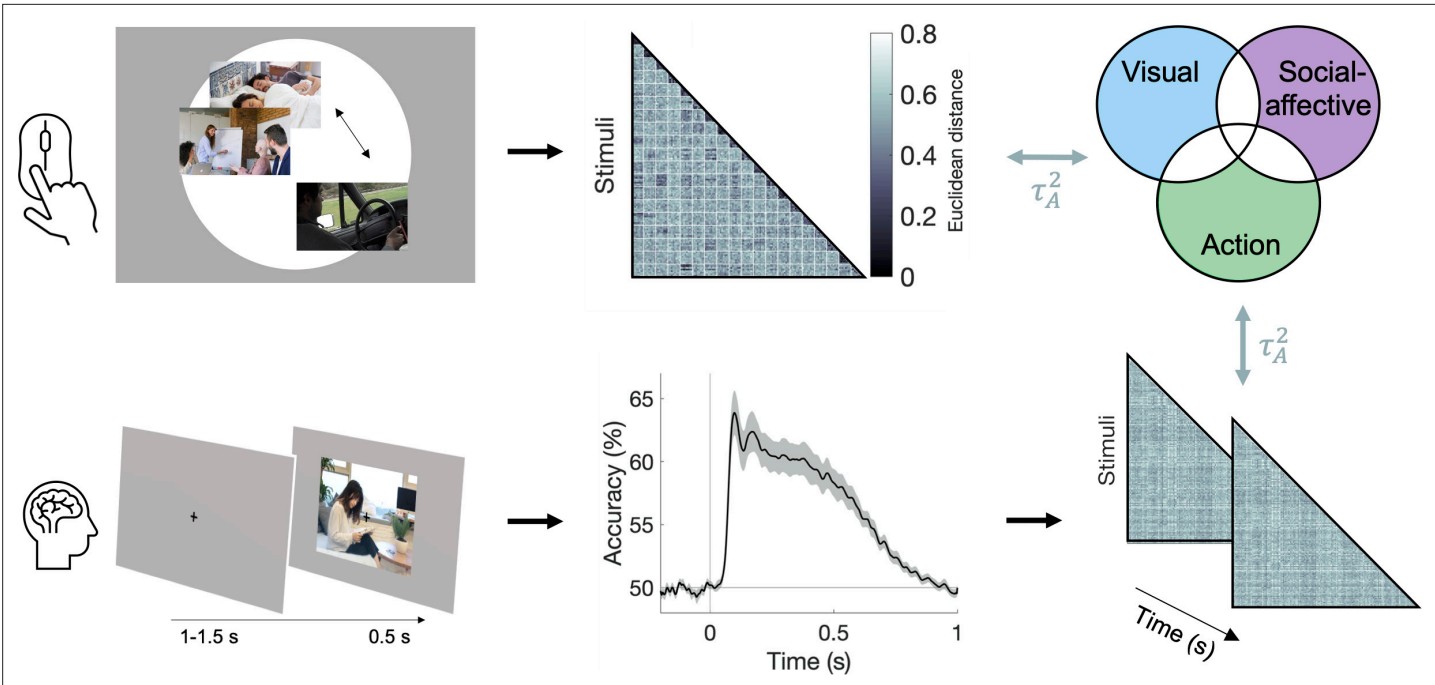

**Figure 2.** Experimental and analysis pipeline for evaluating the contribution of different features to action representations. Above: a multiple arrangement task was used to generate behavioral representational dissimilarity matrices (RDMs) in the two behavioral experiments. Below: electroencephalography (EEG) data was recorded during a one-back task, and time-resolved neural RDMs were generated using pairwise decoding accuracies. Cross-validated variance partitioning was used to assess the unique contributions of visual, social-affective, and action features to the behavioral and neural RDMs, quantified as the predicted squared Kendall's $\tau_A$ . The stimuli in this figure are public domain images similar to the types of videos used in the experiments.

both p<0.001, permutation testing; *Figure 3—figure supplement 1a*). Reliability was significantly higher in Experiment 2 than in Experiment 1 (Mann–Whitney $z$ = 3.21, p=0.0013), potentially reflecting differences in both participant pools and sampling methods (subsets of videos in Experiment 1 vs. full video dataset in Experiment 2; see section 'Multiple arrangement').

We assessed the contribution of 17 different visual, social, and action features to behavior in both experiments by correlating each feature RDM to each participant's behavioral RDM (*Supplementary file 1b*). In Experiment 1 (*Figure 3*), only two visual features were significantly correlated with the behavioral RDMs (environment and activations from the final fully connected layer FC8 of AlexNet). However, there were significant correlations between behavioral RDMs and all action-related RDMs (action category, effectors, transitivity, and activity), as well as all social-affective RDMs (valence, arousal, sociality, and number of agents).

In Experiment 2, the only visual feature that moderately correlated with behavior was the final fully connected layer of AlexNet (p=0.006, below our threshold for significance). Among action features, only effectors and activity were significantly correlated with the behavioral RDMs. However, we found significant correlations with all social-affective features. The results thus converge across both experiments in suggesting that social-affective and, to a lesser extent, action-related features, rather than visual properties, explain behavioral similarity.

## Social-affective features explain the most unique variance in behavioral representations

We performed a cross-validated variance partitioning analysis (*Groen et al., 2018*; *Lescroart et al., 2015*; *Tarhan et al., 2021*) to determine which features contributed the most *unique* variance to behavior (see section 'Variance partitioning'). We selected the 10 features that contributed significantly to behavior in either experiment, that is, two visual features (environment and layer FC8 of AlexNet) and all action and social-affective features. To keep the analysis tractable and understand the contribution of each type of information, we grouped these features according to their type (visual,

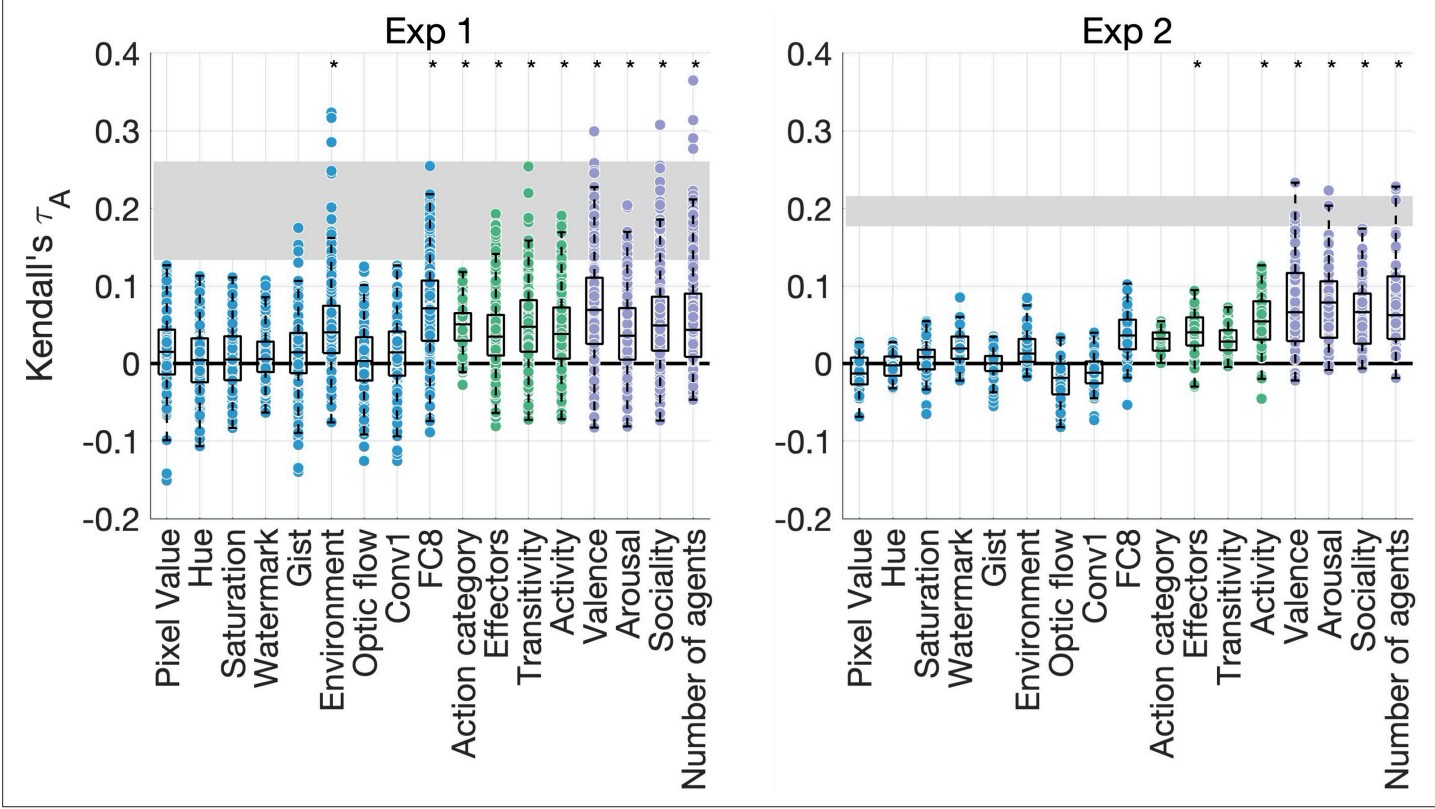

**Figure 3.** Feature contributions to behavioral similarity. Feature-behavior correlations are plotted against the noise ceiling (gray). Each dot is the correlation between an individual participant's behavioral representational dissimilarity matrix (RDM) and each feature RDM. Asterisks denote significance (p<0.005, sign permutation testing). The reliability of the data and feature ratings is presented in *Figure 3—figure supplement 1*.

The online version of this article includes the following figure supplement(s) for figure 3:

**Figure supplement 1.** Reliability of behavioral data.

action, and social-affective) and used them as predictors in a cross-validated hierarchical regression (*Figure 4*). Note that there was no collinearity among the 10 predictors, with an average variance inflation factor of 1.34 (Experiment 1) and 1.37 (Experiment 2).

Together, the 10 predictors explained most of the systematic variance in behavior. In Experiment 1, the predicted squared Kendall's $\tau_A$ of the full model ($\tau_A^2 = 0.06 \pm 0.001$) was higher on average than the true split-half squared correlation ($\tau_A^2 = 0.04 \pm 0.002$). This is likely to be due to the lower reliability of the behavioral similarity data in this experiment and suggests that the 10 predictors are able to explain the data well despite the overall lower prediction accuracy. In Experiment 2, the full model achieved a predicted $\tau_A^2$ of 0.18 ± 0.1 on average compared to a true squared correlation of 0.25 ± 0.1, suggesting that the 10 predictors explain most of the variance (73.21%) in the behavioral data.

In both experiments, social-affective features contributed significantly more unique variance to behavior than visual or action features (*Figure 4*, all Wilcoxon z > 5.5, all p<0.001). While all three groups of features contributed unique variance to behavior in Experiment 1 (all p<0.001, randomization testing), in Experiment 2, only social-affective features contributed significantly to behavior (p<0.001), while visual and action features did not (p=0.06 and 0.47, respectively). Shared variance between feature groups was not a significant contributor in either dataset. Although the effect sizes were relatively low, social-affective features explained more than twice as much unique variance as either the visual or action features in Experiment 1, and six times as much in Experiment 2. Furthermore, given the limits placed on predictivity by the reliability of the behavioral data, affective features predicted a large portion of the explainable variance in both experiments.

The semantic RDM included among the action features was a categorical model based on activity categories (*ATUS, 2019*). To assess whether a more detailed semantic model would explain more variance in behavior, we generated a feature RDM using WordNet similarities between the verb labels

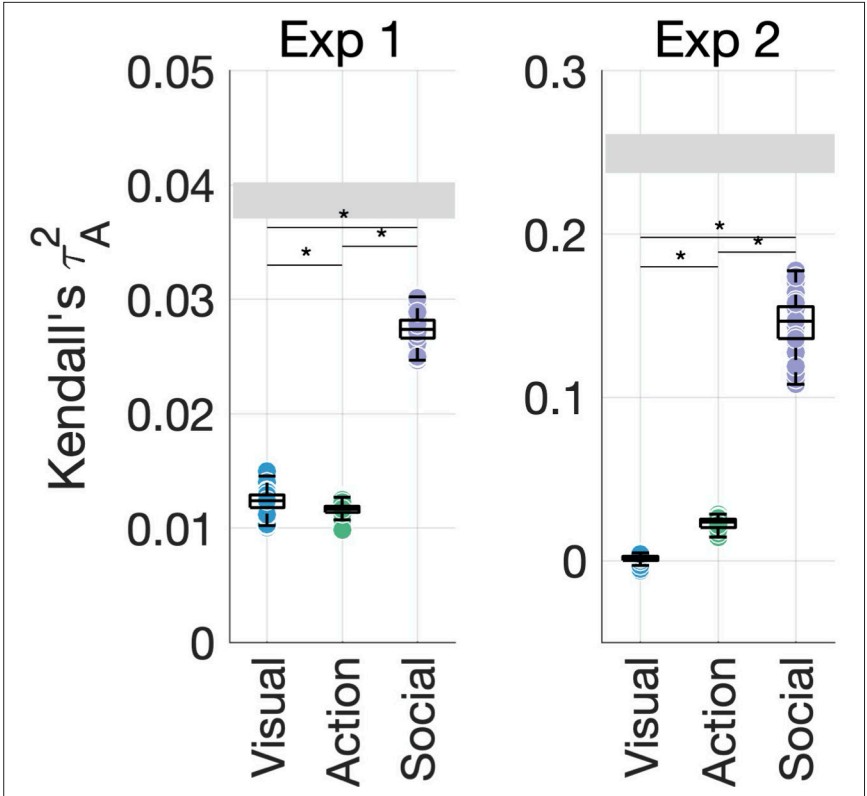

**Figure 4.** Social-affective features explain behavior better than visual and action features. The unique variance explained by visual, action, and social-affective features is plotted against the split-half reliability of the data (gray). Significant differences are marked with asterisks (all p<0.001, Wilcoxon signed-rank tests). For control analyses on how individual features (e.g., action category and the number of agents) and their assignment to groups affect the results, see *Figure 4—figure supplements 1–5*.

The online version of this article includes the following figure supplement(s) for figure 4:

**Figure supplement 1.** Using a more detailed semantic model based on WordNet similarities between video labels does not increase the contribution of action features.

**Figure supplement 2.** Quantifying motion energy as a visual feature does not change the pattern of variance partitioning results.

**Figure supplement 3.** Unique variance explained by visual and action features (environment, FC8, activity, transitivity, effectors, action category) and social-affective features (number of agents, sociality, valence, arousal) in the behavioral data.

**Figure supplement 4.** Unique variance explained by the number of agents, action features (action category, effectors, transitivity, activity), and other social-affective features (sociality, valence, and arousal) in the behavioral data.

**Figure supplement 5.** Unique variance explained by the number of agents, sociality, and affective features (valence and arousal) in the behavioral data.

corresponding to the videos in the Moments in Time dataset. However, replacing the action category RDM with the WordNet RDM did not increase the variance explained by action features (*Figure 4— figure supplement 1*).

Similarly, our decision to quantify motion and image properties separately by using an optic flow model may have reduced the explanatory power of motion features in our data. Indeed, a motion energy model (*Adelson and Bergen, 1985*; *Nunez-Elizalde et al., 2021*) significantly correlated with behavior in Experiment 1, but not in Experiment 2. However, the addition of this model did not change the pattern of unique feature contributions (*Figure 4—figure supplement 2*).

Although the assignment of features to domains was not always straightforward, our results were robust to alternative assignment schemes. For example, high-level visual features can be seen as

bordering the semantic domain, while features like the number of agents or the amount of activity can be seen as visual. However, feature assignment was not the main factor driving our results, which stayed the same even when the activity feature was assigned to the visual group. More strikingly, the social-affective feature group explained significantly more variance than all other features grouped together in both experiments (*Figure 4—figure supplement 3*). This is a particularly stringent test as it pits the unique and shared contributions of all visual, semantic, and action features against the four social-affective features. In Experiment 1, the combined contribution of visual and action features approached that of social-affective features, while in Experiment 2 the difference was larger. Together with the larger contribution of the number of agents in Experiment 2 (*Figure 4—figure supplement 4*, *Figure 4—figure supplement 5*), this suggests that Experiment 2 may have captured more social information, potentially thanks to the exhaustive sampling of the stimuli that allowed each participant to arrange the videos according to different criteria.

Among the social-affective features we tested, the number of agents could be seen as straddling the visual and social domains. To assess whether our results were driven by this feature, we performed a control variance partitioning analysis pitting the number of agents against the other, higher-level social-affective features (*Figure 4—figure supplement 3*). In both experiments, the higher-level features (sociality, valence, and arousal) contributed more unique variance than the number of agents, suggesting that our results are not explained by purely visual factors.

Furthermore, an additional analysis looking at the separate contributions of the number of agents, sociality, and affective features (valence and arousal) found that the affective features contributed the greatest variance in both experiments (*Figure 4—figure supplement 5*). For technical reasons, this analysis compared the joint contribution of both affective features to each single social feature and did not discount the impact of variance shared with visual or action-related features. Despite these limitations, the results suggest that the contribution of the social-affective feature group is not driven by the number of agents or the variance it shares with sociality, and highlight the role of affective features (valence and arousal) in explaining behavior.

## EEG patterns reflect behavioral similarity

We performed an EEG experiment to investigate how action-relevant features are processed over time. Participants viewed 500 ms segments of the 152 videos from Experiment 1 and performed a

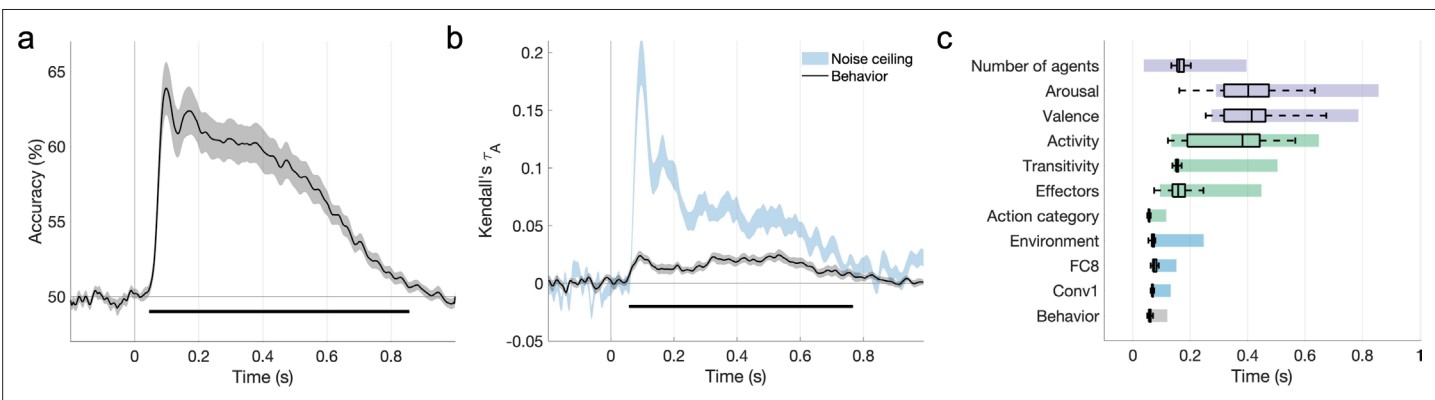

**Figure 5.** The features that explain behavioral action representations also contribute to neural representations. (**a**) Time course of video decoding accuracy, averaged across all pairs of videos and participants (in gray: SEM across participants). The horizontal line marks above-chance performance (sign permutation testing, cluster-corrected $p<0.05$). (**b**) Behavioral similarity correlates with the neural representational dissimilarity matrices (RDMs). The noise ceiling is shown in light blue (leave-one-subject-out correlation, mean ± SEM). Horizontal lines mark significant time windows (sign permutation testing, cluster-corrected $p<0.05$). (**c**) The distribution of significant correlation onsets for each feature model across 1000 bootstrapping iterations (sign permutation testing, cluster-corrected $p<0.05$). Color rectangles show 90% confidence intervals. The time courses of all feature correlations are shown in *Figure 5—figure supplement 1*. The average electroencephalography (EEG) evoked response is visualized in *Figure 5—figure supplement 2*.

The online version of this article includes the following figure supplement(s) for figure 5:

**Figure supplement 1.** Correlations between features and the time-resolved neural representational dissimilarity matrices (RDMs).

**Figure supplement 2.** Average electroencephalography (EEG) evoked response.

one-back action task in which they detected repetitions of the action category (see section 'EEG: Experimental procedure'). To relate neural patterns to behavioral and feature RDMs, we computed time-resolved neural RDMs for each participant using decoding accuracies between all pairs of videos (*Figures 2 and 5a*). The time course of decoding performance was similar to that observed in previous E/MEG studies using still visual stimuli (*Carlson et al., 2013*; *Cichy et al., 2014*; *Dima et al., 2018*; *Greene and Hansen, 2018*; *Isik et al., 2014*). Decoding accuracy rose above chance at 50 ms after video onset, reached its maximum at 98 ms (63.88 ± 6.82% accuracy), and remained above chance until 852 ms after video onset (cluster-corrected p<0.05, sign permutation testing).

To assess brain–behavior correlations, we related the average behavioral RDM obtained in Experiment 1 to the time-resolved neural RDMs (Kendall's $\tau_A$). The behavioral RDM correlated significantly with neural patterns during a cluster between 62 and 766 ms after video onset (*Figure 5b*), suggesting that the features guiding the intuitive categorization of naturalistic actions also underlie their neural organization.

## Neural timescale of individual feature representations

We assessed the correlations between EEG patterns and the 10 feature RDMs found to contribute to behavior in Experiment 1. We also included an additional feature RDM based on the first convolutional layer of AlexNet, which best captures early visual neural responses (*Figure 5—figure supplement 1*; see section 'Multivariate analysis'). The feature RDMs that contributed to behavioral similarity also correlated with the EEG patterns (*Figure 5—figure supplement 1*), with a single exception (sociality).

A bootstrapping analysis of the cluster onsets of these correlations (*Figure 5c*) suggests a progression from visual to action and social-affective features. Visual predictors correlated with the neural patterns between 65 ± 15ms (mean ± SD, Conv1) and 84 ± 62 ms (Environment), while action category also had an early onset (58 ± 9 ms). Other action-related features, however, emerged later (transitivity: 170 ± 67 ms, effectors: 192 ± 94 ms, activity: 345 ± 133 ms). Among social-affective features, the number of agents had the earliest correlation onset (178 ± 81 ms), while valence and arousal emerged later (395 ± 81 and 404 ± 91 ms, respectively). Importantly, these features are spontaneously extracted in the brain, as none of them, with the exception of action category, were directly probed in the one-back task performed by participants. In addition, all features were extracted during behaviorally relevant time windows (*Figure 5b*).

## A temporal hierarchy in action perception

A cross-validated variance partitioning analysis revealed different stages in the processing of naturalistic actions (*Figure 6*). Visual features dominated the early time windows (66–138 ms after video onset). Action features also contributed a significant amount of unique variance (162–598 ms), as well as variance shared with social-affective features (354–598 ms; *Figure 6—figure supplement 1*). Finally, social-affective features independently predicted late neural responses (446–782 ms). Importantly, visual features did not share a significant amount of variance with either action or social-affective features.

An analysis of effect onsets across 100 split-half iterations points to the hierarchical processing of these features, with a progression from visual to action to social-affective features. Social-affective features (mean onset 418 ± 89 ms) contributed unique variance significantly later than other feature sets, while action features (245 ± 104 ms) came online later than visual features (65 ± 8ms; all Wilcoxon z > 7.27, p<0.001; *Figure 6b*). A fixed-effects analysis revealed the same order of feature information with larger effect sizes (*Figure 6—figure supplement 2*).

Motion has been shown to drive the response of visual areas to naturalistic stimuli (*Russ and Leopold, 2015*; *Nishimoto et al., 2011*). To better assess the effect of motion on EEG responses, we performed an additional analysis including the motion energy model. There was a sustained correlation between motion energy and EEG patterns beginning at 62 ms (*Figure 6—figure supplement 3*). In the variance partitioning analysis, the addition of motion energy increased the unique contribution of visual features and decreased that of action features, indicating that the action features share variance with motion energy. However, the three stages of temporal processing were preserved in the fixed-effects analysis even with the addition of motion energy, suggesting that the three feature groups made distinct contributions to the neural patterns. Importantly, the unique contribution of social-affective features was unchanged in both analyses by the addition of the motion energy model.

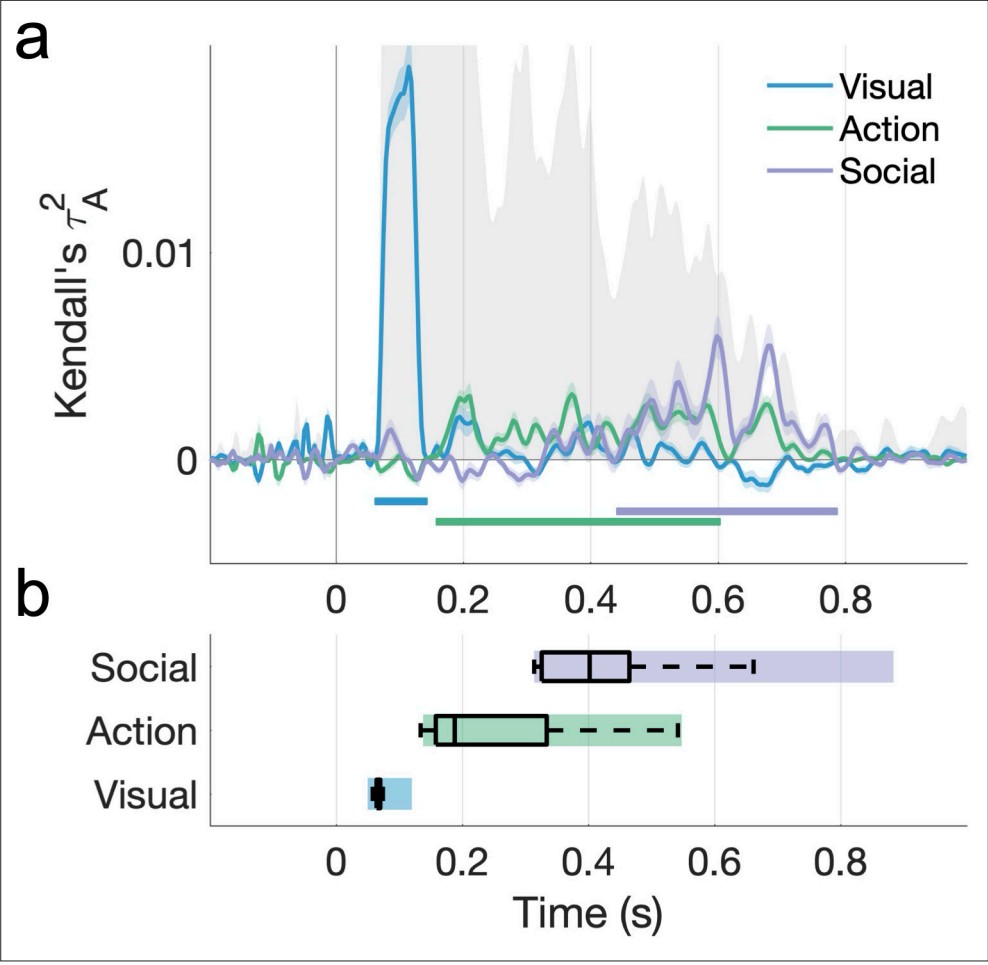

**Figure 6.** Hierarchical processing of visual, action, and social-affective features. (**a**) Unique variance explained by each group of features over time. The split-half reliability of the data is shown in gray (shaded area; see also *Figure 5b*). Horizontal lines mark significant time windows (sign permutation testing, cluster-corrected p<0.05). The time course of shared variance is displayed in *Figure 6—figure supplement 1*. See *Figure 6—figure supplement 2* for the results of a fixed-effects analysis. *Figure 6—figure supplement 3* shows how the addition of a motion energy model affects these results. (**b**) The distribution of effect onsets across 100 split-half iterations (sign permutation testing, cluster-corrected p<0.05). Color rectangles show 90% confidence intervals.

The online version of this article includes the following figure supplement(s) for figure 6:

**Figure supplement 1.** Shared variance among visual, action, and social predictors in the cross-validated variance partitioning analysis.

**Figure supplement 2.** Fixed-effects variance partitioning results (stacked area plot).

**Figure supplement 3.** The contribution of motion energy to the neural data.

## Discussion

Here, we used a large-scale naturalistic stimulus set to disentangle the roles of different features in action perception. Two novel findings emerge from our study. First, our behavioral results suggest that social-affective features play the most important role in how we organize naturalistic everyday actions, above and beyond fundamental visual and action features like scene setting or action category. Second, these behaviorally relevant features are spontaneously extracted in the brain and follow a hierarchical sequence from visual to action-related and culminating with social-affective features. These results offer an account of how internal representations of everyday actions emerge in the mind and brain.

## Behavioral representations: What features support action perception?

Across two separate multiple arrangement experiments with large-scale naturalistic stimulus sets, we found that social-affective features predicted similarity judgments better than, and independently of, visual and action-related features. By sampling a comprehensive feature space ranging from low-level to conceptual, we were able to distinguish between components that often covary, such as scene setting and action category or sociality and transitivity. Previous studies have operationalized features in different ways, and an exhaustive investigation is thus difficult; however, our approach of including several important features from each group mitigated this, as suggested by the high amount of variance in behavior collectively explained by our features.

Our work adds to a growing body of evidence for the importance of social-affective features in action perception and extends it by disentangling the contributions of specific social and semantic features. Previous work has highlighted sociality as an essential feature in neural action representations (*Tarhan and Konkle, 2020*; *Wurm et al., 2017*; *Wurm and Caramazza, 2019*), and a recent study (*Tarhan et al., 2021*) found that behavioral action similarity judgments were better explained by similarity in actors' goals than by visual similarity. In line with this work, we found a minimal contribution of visual features to action similarity judgments. In contrast, all of our social-affective features – the number of agents, sociality, valence, and arousal – were significantly correlated with behavioral similarity. Furthermore, only two individual action-related features replicated across the two experiments: the amount of activity and the effector (body part) feature, the latter of which is highly relevant to the actors' goals. This could be interpreted as further evidence for the importance of socially relevant features in our internal representations of actions, and identifies specific social and goal-related features that are important for action understanding.

A hypothesis-driven approach will always pose challenges due to practical limitations in the number of feature spaces one can feasibly test. Our approach of grouping predictors together based on theoretical distinctions made it possible to rigorously evaluate the unique contributions of different types of features, which is an essential first step in understanding naturalistic action representations. This analysis revealed that social-affective features contributed the most unique variance in both experiments, suggesting that they robustly predict behavioral similarity judgments, while visual and action features explained little unique variance in either experiment (*Figure 4*). An exploratory follow-up analysis showed that this effect was primarily driven by affective features (valence and arousal), with the number of agents as a secondary contributor. Recent work found that affective features drive the perceived similarity of memories of real-life events (*Tomita et al., 2021*), suggesting that these features bridge the action, event, and memory domains in organizing mental representations.

Among our social-affective features, the number of agents could be construed as a perceptual precursor to sociality. Indeed, previous fMRI work has suggested that neural representations of actions in the visual system reflect perceptual precursors of social features rather than higher-level social features (*Wurm and Caramazza, 2019*). Here, we found that high-level social-affective features (particularly valence and arousal) contributed significantly to behavior independently of the number of agents. Further, affective features explained significantly more unique variance in behavior than the number of agents in both experiments (*Figure 4—figure supplements 4 and 5*). Our findings suggest that high-level social-affective features, and in particular valence and arousal, uniquely drive human action representations.

## Neural representations: How does action perception unfold over time?

Using EEG, we tracked the temporal dynamics of naturalistic action perception. Using naturalistic stimuli and a rich feature space enabled us to disentangle the contributions of different features and investigate their relative timing. Visual, action, and social-affective features made unique contributions to the EEG patterns at different processing stages, revealing a representational hierarchy of spontaneously extracted features.

Almost all behaviorally relevant features correlated with the EEG patterns, with action-related and social-affective features emerging later than visual features (*Figure 5c*). Most action-related features emerged within 200 ms, on the timescale of feedforward processing, which is consistent with prior work showing invariant responses to actions as early as 200 ms (*Isik et al., 2018*; *Tucciarelli et al., 2015*), and action transitivity processing as early as 250 ms (*Wamain et al., 2014*). Among social-affective features, the number of agents emerged earliest (162 ms), pointing to the role of this feature

as a perceptual precursor in social perception (*Papeo, 2020*; *Wurm and Caramazza, 2019*). Valence and arousal emerged later, around 400 ms after video onset. Interestingly, sociality, which has been highlighted as an important dimension in previous fMRI work on action perception (*Tarhan and Konkle, 2020*; *Wurm et al., 2017*), did not correlate with the EEG patterns. This effect was not likely to be driven by a lower reliability in the measurement of this feature, as sociality was more reliable than all other behaviorally rated features in Experiment 1 (*Figure 3—figure supplement 1*). While the absence of an effect does not preclude the possibility that this feature is being processed, it is possible that prior work has confounded sociality with other correlated social-affective features (such as the number of agents or arousal). Alternatively, our operationalization of this feature (which was broader than in some previous studies, e.g., *Tucciarelli et al., 2019*; *Wurm et al., 2017*) may have led to differences in the information captured. Note that this finding is mirrored in our behavioral results, where we observed larger unique contributions from valence, arousal, and the number of agents than sociality (*Figure 4—figure supplement 5*).

Importantly, these features emerged spontaneously as the one-back task performed during the EEG recordings only related to action category. However, the semantic processing required to perform the task may have contributed to these computations. The emergence of features irrelevant to the task at hand (action category is not correlated with any other features in the dataset) suggests that this temporal hierarchy would also emerge in the absence of a task; however, future work can more directly test the impact of implicit and explicit (e.g., social-affective) processing on these neural dynamics.

Variance partitioning revealed a clear temporal progression from visual features (~100 ms) to action features (~150–600 ms) to social-affective features (~400–00 ms). Importantly, these processing stages emerged after partialling out the contributions of other groups of predictors in a cross-validated analysis, validating our a priori distinctions between feature classes. These findings suggest that the extraction of visual features occurs rapidly, within 200 ms, and is likely supported by feedforward computations. The social-affective features that support behavioral representations, however, were extracted last. This is consistent with theories suggesting that internal visual experience reverses the stages of perceptual processing (*Dijkstra et al., 2020*; *Hochstein and Ahissar, 2002*). Specifically, it was the final, social-affective stage of neural processing that was reflected in the intuitive behavioral representations, and not the initially extracted visual features. Furthermore, action-related features were extracted significantly before social-affective features, suggesting the two are not extracted in parallel, but instead pointing to a hierarchy in which both visual and action-related features may contribute to socially relevant computations. Given the short duration of our videos and the relatively long timescale of neural feature processing, it is possible that social-affective features are the result of ongoing processing relying on temporal integration of the previously extracted features. However, more research is needed to understand how these temporal dynamics change with continuous visual input (e.g., a natural movie), and whether social-affective features rely on previously extracted information.

Our results add temporal characterization to previous fMRI findings, suggesting that the seemingly conflicting features revealed by previous studies, like the number of agents (*Wurm and Caramazza, 2019*), sociality (*Tarhan and Konkle, 2020*), or semantic action category (*Tucciarelli et al., 2019*), emerge at different stages during action observation. Thus, the existence of different organizing dimensions can be explained not just through spatial segregation within and across brain areas, but also through a temporal gradient starting with visual features and concluding with behaviorally relevant social and affective representations. More work is needed to understand where these dynamic representations emerge in the brain, and whether they are supported by overlapping or distinct networks. Future research could test this using EEG-fMRI fusion to track the spatiotemporal dynamics of action representations.

## Actions in context

As real-world actions tend to occur in a rich social context, studies of action perception should consider socially relevant features and the interactions between different systems for perceiving actions, agents, and their mental states (*Quadflieg and Koldewyn, 2017*). Recent work suggests that social perception enhances visual processing (*Bellot et al., 2021*; *Papeo, 2020*) and recruits dedicated neural circuits (*Isik et al., 2017*; *Pitcher and Ungerleider, 2021*). Our findings open exciting

new avenues for connecting these areas of research. For example, future studies could more explicitly disentangle the perceptual and conceptual building blocks of social and affective features, such as body posture or facial expression, and their roles in action and interaction perception.

One fundamental question that lies at the root of this work is how actions should be defined and studied. Here, we adopted a broad definition of the term, focusing on activities as described in the ATUS (*ATUS, 2019*). Although our stimuli were selected to clearly depict short, continuous actions performed by visible agents, their naturalistic and context-rich nature means that they could be understood as 'events,' encompassing elements that are not singularly specific to actions. A wealth of evidence has shown that context changes visual processing in a nonadditive way (*Bar, 2004*; *Willems and Peelen, 2021*), and emerging evidence suggests that the same is true for actions (*Wurm et al., 2012*). Studying actions in context holds promise for understanding how semantically rich representations emerge in naturalistic vision. This, in turn, will pave the way towards a computational understanding of the neural processes that link perception and cognition.

## Materials and methods
### Behavior: Stimuli

We curated two stimulus sets containing three-second videos of everyday actions from the Moments in Time dataset (*Monfort et al., 2020*). To broadly sample the space of everyday actions, we first identified the most common activities from the National Bureau of Labor Statistics' American Time Use Survey (*ATUS, 2019*). Our final dataset included 18 social and nonsocial activities that lend themselves to visual representation (*Table 1*), to ensure a diverse and balanced stimulus set representative of the human everyday action space. We note that the ATUS distinctions are based on performed rather than observed actions. While imperfect, they provide ecologically relevant and objective criteria with which to define our action space.

Action categories were selected from the second-level activities identified in the ATUS. We used a minimum cutoff of 0.14 hr/day to select common actions (*Table 1*). To diversify our dataset, we added a 'hiking' category (to increase variability in scene setting) and a 'fighting' category (for variability along affective dimensions). In addition, 'driving' was selected as a more specific instance of the 'travel' categories in ATUS as it is the most common form of transportation in the United States. Some adjustments were also made to the 'relaxing and leisure' category by selecting two specific activities that were easy to represent and distinguish visually, as well as above threshold ('reading' and 'playing games'). In addition, our 'reading' category included both 'reading for personal interest' and 'homework and research' as this distinction is difficult to convey visually. We omitted three leisure categories that were difficult to represent in brief videos ('watching TV,' 'relaxing and thinking,' and 'computer use for leisure, excluding games'), as well as the 'consumer goods purchases' category.

We curated an initial set of 544 videos from the Moments in Time dataset by identifying the verb labels relevant to our chosen activities. Videos were chosen that were horizontally oriented (landscape), of reasonable image quality, clearly represented the activity in question, clearly depicted an agent performing the activity, and varied in terms of number of agents, gender and ethnicity of agents, and scene setting. Although some categories were represented by fewer verb labels than others in the final set, our curation procedure aimed to balance important features within and across action categories. We also curated a set of control videos depicting natural and indoors scenes.

We then selected two subsets of videos (1) that sampled all activities in a balanced manner and (2) where sociality (as assessed through behavioral ratings, see section 'Behavioral ratings') was minimally correlated to the number of agents (experimenter-labeled). This was done by randomly drawing 10,000 subsets of videos that sampled all activities equally and selecting the video set with the lowest correlation between sociality and the number of agents. These two features are difficult to disentangle in naturalistic stimulus sets, and we were able to minimize, though not fully eliminate, this correlation (*Figure 1a*).

The first stimulus set contained 152 videos (eight videos per activity and eight additional videos with no agents) and was used in Experiment 1. The videos with no agents were included to provide variation in the dataset along visual properties that did not pertain to actions or agents, as well as variation in the overall dataset in terms of number of agents per video. From the remaining videos, a second set of 76 videos was sampled and manually adjusted to remove videos without agents (in the

interest of experimental time) and any videos that were too similar visually to other videos in the same category (e.g., involving a similar number of agents in similar postures). The second stimulus set thus contained 65 videos (three or four videos per activity) and was used in Experiment 2. The videos were preprocessed to a frame rate of 24 frames per second using the *videoWriter* object in MATLAB and resized to 600 × 400 pixels. This was done by first resizing the videos in order to meet the dimension criteria (using MATLAB's *imresize* function with bicubic interpolation). The videos were then cropped to the correct aspect ratio either centrally or using manually determined coordinates to make sure the action remained clear after cropping.

## Behavior: Participants

### Behavioral ratings

A total of 256 workers (202 after exclusions, located in the United States, worker age and gender not recorded) from the online platform Amazon Mechanical Turk provided sociality, valence, arousal, and activity ratings of the video stimuli, and 43 workers (35 after exclusions) provided transitivity ratings.

### Multiple arrangement

Two separate online multiple arrangement experiments were performed on each of the two stimulus sets. A total of 374 workers from Amazon Mechanical Turk took part in Experiment 1 (300 after exclusions, located in the United States, worker age and gender not recorded). Experiment 2 involved 58 participants (53 after exclusions, 31 female, 20 male, 1 non-binary, 1 not reported, mean age 19.38 ± 1.09) recruited through the Department of Psychological and Brain Sciences Research Portal at Johns Hopkins University.

All procedures for online data collection were approved by the Johns Hopkins University Institutional Review Board (protocol number HIRB00009730), and informed consent was obtained from all participants.

## Behavior: Experimental procedure

### Behavioral ratings

Participants viewed subsets of 30–60 videos from the initially curated large-scale set and rated the events depicted on a five-point scale. In a first set of experiments, the dimensions rated were sociality (how social the events were, from 1 – not at all to 5 – very social); valence (how pleasant the events were, from 1 – very unpleasant to 5 – very pleasant); arousal (how intense the events were, from 1 – very calm to 5 – very intense); and activity (how active they were, from 1 – no action to 5 – very active). In separate experiments, participants provided transitivity ratings for the two final stimulus sets (i.e., to what extent the actions involved a person or people interacting with an object, from 1 – not at all to 5 – very much). Participants were excluded if they responded incorrectly to catch trials (approximately 10% of trials) requiring them to label the action shown in the prior video, or if they provided overly repetitive ratings (e.g., using only two unique values or fewer out of five possible ratings throughout the entire experiment). This amounted to an average of 17.46 ± 2.14 ratings per video (Experiment 1) and 18.22 ± 2.09 ratings per video (Experiment 2). The experiments were implemented in JavaScript using the jsPsych library (*de Leeuw, 2015*).

### Multiple arrangement

To characterize human action representations, we measured behavioral similarity using two multiple arrangement experiments. The experiments were conducted on the Meadows platform (https://meadows-research.com/) and required participants to arrange the videos according to their similarity inside a circular arena. Participants were free to use their own criteria to determine similarity, so as to encourage natural behavior.

Each trial started with the videos arranged around the circular arena. The videos would start playing on hover, and the trial would not end until all videos were played and dragged-and-dropped inside the arena (*Figure 2*). Different sets of videos were presented in different trials. An adaptive 'lift-the-weakest' algorithm was used to resample the video pairs placed closest together, so as to gather sufficient evidence (or improve the signal-to-noise ratio) for each pair. This procedure was repeated until an evidence criterion of 0.5 was reached for each pair or until the experiment timed out (Experiment 1: 90 min; Experiment 2: 120 min). By asking participants to zoom into the subsets previously judged

as similar, the task required the use of different contexts and criteria to judge relative similarities. Compared to other methods of measuring similarity, multiple arrangement thus combines efficient sampling of a large stimulus set with adaptive behavior that can recover a multi-dimensional similarity structure (*Kriegeskorte and Mur, 2012*).

In Experiment 1, participants arranged different subsets of 30 videos from the 152-video set, with a maximum of 7 videos shown in any one trial. The stimuli were sampled in a balanced manner across participants. The task took on average 32 ± 14.4 min and 86.8 ± 22.6 trials.

In Experiment 2, all participants arranged the same 65 videos (entire 65-video set), with a maximum of 8 videos shown in any one trial. The task took on average 87.5 ± 24.6 min, including breaks, and 289.7 ± 57.3 trials.

The experiments included a training trial in which participants arranged the same seven videos (in Experiment 1) or eight videos (in Experiment 2) before beginning the main task. In both experiments, these videos were hand-selected to represent clear examples from four categories. Participants were excluded from further analysis if there was a low correlation between their training data and the average of all other participants' data (over 2 SDs below the mean). They were also excluded if they responded incorrectly to a catch trial requiring them to label the action in previously seen videos.

Inverse MDS was used to construct behavioral dissimilarity matrices containing normalized Euclidean distances between all pairs of videos (*Kriegeskorte and Mur, 2012*). In Experiment 1, the behavioral RDM contained 11,476 pairs with an average of 11.37 ± 3.08 estimates per pair; in Experiment 2, there were 2080 pairs arranged by all 53 participants.

## Behavior: Data analysis
### Representational similarity analysis

Everyday actions can be differentiated along numerous axes. Perceptually, they can differ in terms of visual properties, like the setting in which they take place. They can also be characterized through action-related features, like semantic action category, or through social features, like the number of agents involved. Understanding how these features contribute to natural behavior can shed light on how naturalistic action representations are organized. Here, we used representational similarity analysis (RSA) to assess the contribution of visual, action, and social-affective features to the behavioral similarity data.

We quantified features of interest using image properties, labels assigned by experimenters (*Supplementary file 1a*), and behavioral ratings (provided by participants, see section 'Behavioral ratings'). We calculated the Euclidean distances between all pairs of stimuli in each feature space, thus generating 17 feature RDMs.

To quantify visual features, image properties were extracted separately for each frame of each video and averaged across frames. These included pixel value (luminance), hue, saturation, optic flow (the magnitude of the optic flow estimated using the Horn–Schunck method), and the spatial envelope of each image quantified using GIST (*Oliva and Torralba, 2001*). We also extracted activations from the first convolutional layer and last fully-connected layer of a pretrained feedforward convolutional neural network (AlexNet; *Krizhevsky et al., 2012*). These features were vectorized prior to computing Euclidean distances between them (see *Supplementary file 1b* for the dimensionality of each feature). Two additional experimenter-labeled features were included: scene setting (indoors/outdoors) and the presence of a watermark. To assess whether a motion energy model (*Adelson and Bergen, 1985*; *Nishimoto et al., 2011*; *Watson and Ahumada, 1985*) would better capture the impact of motion, we performed control analyses by computing motion energy features for each video using a pyramid of spatio-temporal Gabor filters with the *pymoten* package (*Nunez-Elizalde et al., 2021*).

Action feature RDMs were based on transitivity and activity ratings (provided by participants, see above), as well as action category (a binary RDM clustering the stimuli into activity categories based on the initial dataset designations) and effectors (experimenter-labeled). The latter consisted of binary vectors indicating the involvement of body parts in each action (face/head, hands, arms, legs, and torso). To assess whether a more detailed semantic model would capture more information, we also performed a control analysis using a feature RDM based on WordNet similarities between the verb labels in the 'Moments in Time' dataset (*Figure 4—figure supplement 1*).

Social-affective feature RDMs were based on sociality, valence, and arousal ratings (all provided by participants, see section 'Behavioral ratings' above) and the number of agents in each video, which was labeled by experimenters on a four-point scale (from 0, no agent present, to 3, three or more agents present).

Each participant's behavioral RDM was correlated to the feature RDMs, and the resulting Kendall's $\tau_A$ values were tested against chance using one-tailed sign permutation testing (5000 iterations). P-values were omnibus-corrected for multiple comparisons using a maximum correlation threshold across all models (*Nichols and Holmes, 2002*).

A noise ceiling was calculated by correlating each subject's RDM to the average RDM (upper bound), as well as to the average RDM excluding the left-out subject (lower bound; *Nili et al., 2014*).

### Variance partitioning

Despite low correlations between features of interest in both stimulus sets (*Figure 1a*), shared variance could still contribute to the RSA results. To estimate the unique contributions of the three primary groups of features, we performed a cross-validated variance partitioning analysis, excluding individual features that did not correlate with the behavioral data in the above RSA analysis. The three groups included visual features (scene setting and the last fully connected layer of AlexNet), action features (action category, effectors, transitivity, action), and social-affective features (number of agents, sociality, valence, arousal).

The behavioral data were randomly split into training and test sets (100 iterations) by leaving out half of the individual similarity estimates for each pair of videos in Experiment 1 (since different participants saw different subsets of videos) or half of the participants in Experiment 2. We fit seven different regression models using the average training RDM (with every possible combination of the three groups of features), and we calculated the squared Kendall's $\tau_A$ between the predicted responses and the average test RDM. These values were then used to calculate the unique and shared portions of variance contributed by the predictors (*Groen et al., 2018*; *Lescroart et al., 2015*; *Tarhan et al., 2021*).

The resulting values were tested against chance using one-tailed sign permutation testing (5000 iterations, omnibus-corrected for multiple comparisons). Differences between groups of features were assessed with two-sided Wilcoxon signed-rank tests.

## EEG: Stimuli

The stimulus set from behavioral Experiment 1 was used in the EEG experiment, containing 152 videos from 18 categories, as well as control videos. The three-second stimuli were trimmed to a duration of 0.5 s centered around the action as determined by visual inspection, to ensure that the shorter videos were easily understandable. This helped improve time-locking to the EEG signals and allowed for a condition-rich experimental design. An additional 50 videos were included as catch stimuli (25 easily identifiable pairs depicting the same action, manually chosen from the larger stimulus set).

## EEG: Participants

Fifteen participants (six female, nine male, mean age 25.13 ± 6.81) took part in the EEG experiment. All participants were right-handed and had normal or corrected-to-normal vision. Informed consent was obtained in accordance with the Declaration of Helsinki, and all procedures were approved by the Johns Hopkins University Institutional Review Board (protocol number HIRB00009835).

## EEG: Experimental procedure

Continuous EEG recordings with a sampling rate of 1000 Hz were made with a 64-channel Brain Products ActiCHamp system using actiCAP electrode caps in a Faraday chamber. Electrode impedances were kept below 25 kΩ when possible, and the Cz electrode was used as an online reference.

Participants were seated upright while viewing the videos on a back-projector screen situated approximately 45 cm away. The 152 videos were shown in pseudorandom order in each of 10 blocks with no consecutive repetition allowed. In addition, four repetitions of the 25 catch video pairs were presented at random times during the experiment. The video pairs were presented in different orders to minimize learning effects, so that for each video pair (V1, V2), half of the presentations were in the order V1-V2 and half of them were in the order V2-V1. Participants performed a one-back task

and were asked to press a button on a Logitech game controller when they detected two consecutive videos showing the same action. Participants were not instructed on what constituted an action, beyond being given 'eating' as a simple example. There was a break every 150 trials, and participants could continue the experiment by pressing a button. In total, the experiment consisted of 1720 trials (1520 experimental trials and 200 catch trials) and took approximately 45 min.

The stimuli were presented using an Epson PowerLite Home Cinema 3000 projector with a 60 Hz refresh rate. Each trial started with a black fixation cross presented on a gray screen for a duration chosen from a uniform distribution between 1 and 1.5 s, followed by a 0.5 s video. The stimuli were presented on the same gray background and subtended approximately 15 × 13 degrees of visual angle. The fixation cross remained on screen, and participants were asked to fixate throughout the experiment. A photodiode was used to accurately track on-screen stimulus presentation times and account for projector lag. The paradigm was implemented in MATLAB R2019a using the Psychophysics Toolbox (*Brainard, 1997*; *Kleiner et al., 2007*; *Pelli, 1997*).

## EEG: Data analysis

### Preprocessing

EEG data preprocessing was performed using MATLAB R2020b and the FieldTrip toolbox (*Oostenveld et al., 2011*). First, the EEG data were aligned to stimulus onset using the photodiode data to correct for any lag between stimulus triggers and on-screen presentation. The aligned data were segmented into 1.2 s epochs (0.2 s pre-stimulus to 1 s post-stimulus onset), baseline-corrected using the 0.2 s prior to stimulus onset, and high-pass filtered at 0.1 Hz.

Artifact rejection was performed using a semi-automated pipeline. First, the data were filtered between 110 and 140 Hz and Hilbert-transformed to detect muscle artifacts; segments with a z-value cutoff above 15 were removed. Next, channels and trials with high variance were manually rejected based on visual inspection of a summary plot generated using the *ft_rejectvisual* function in FieldTrip. Finally, independent component analysis (ICA) was performed to identify and remove eye movement components from the data.

Catch trials were removed from the data together with any trials that elicited a button response (13.74 ± 1.82% of all trials). Of the remaining trials, 8.36 ± 5.01% (ranging between 25 and 275 trials) were removed during the artifact rejection procedure. A maximum of two noisy electrodes were removed from eight participants' datasets.

Prior to further analysis, the data were re-referenced to the median across all electrodes, low-pass filtered at 30 Hz to investigate evoked responses, and downsampled to 500 Hz.

### Multivariate analysis

We performed multivariate analyses to investigate (1) whether EEG patterns reflected behavioral similarity and (2) whether different visual, action, and social-affective features explained variance in the neural data.

First, time-resolved decoding of every pair of videos was performed using a linear support vector machine classifier as implemented in the LibSVM library (*Chang and Lin, 2011*). Split-half cross-validation was used to classify each pair of videos in each participant's data. To do this, the single-trial data was divided into two halves for training and testing, whilst ensuring that each condition was represented equally. To improve SNR, we combined multiple trials corresponding to the same video into pseudotrials via averaging. The creation of pseudotrials was performed separately within the training and test sets. As each video was shown 10 times, this resulted in a maximum of five trials being averaged to create a pseudotrial. Multivariate noise normalization was performed using the covariance matrix of the training data (*Guggenmos et al., 2018*). Classification between all pairs of videos was performed separately for each time point. Data were sampled at 500 Hz, and so each time point corresponded to nonoverlapping 2 ms of data. Voltage values from all EEG channels were entered as features to the classification model.

The entire procedure, from dataset splitting to classification, was repeated 10 times with different data splits. The average decoding accuracies between all pairs of videos were then used to generate a neural RDM at each time point for each participant. To generate the RDM, the dissimilarity between each pair of videos was determined by their decoding accuracy (increased accuracy representing increased dissimilarity at that time point).

Next, we evaluated the correlations between each participant's time-resolved neural RDM and the feature RDMs found to correlate with behavioral similarity (Experiment 1). To investigate the link between behavioral and neural representations, we also correlated neural RDMs with the average behavioral RDM obtained from the multiple arrangement task in Experiment 1. This analysis was performed using 10 ms sliding windows with an overlap of 6 ms. The resulting Kendall's $\tau_A$ values were tested against chance using one-tailed sign permutation testing (5000 iterations, cluster-corrected for multiple comparisons across time using the maximum cluster sum, $\alpha = 0.05$, cluster setting $\alpha = 0.05$). A noise ceiling was calculated using the same procedure as in the behavioral RSA (see 'Representational similarity analysis'). Effect latencies were assessed by bootstrapping the individual correlations 1000 times with replacement to calculate 90% confidence intervals around effect onsets.

To quantify the contributions of visual, social-affective, and action features to the neural RDMs, a time-resolved cross-validated variance partitioning procedure was performed. Using 100 split-half cross-validation iterations, the neural RDM was entered as a response variable in a hierarchical regression with three groups of feature RDMs (visual, social-affective, and action) as predictors. This analysis employed the same 10 feature RDMs used in the behavioral variance partitioning (see section 'Variance partitioning'), with the addition of activations from the first convolutional layer of AlexNet (Conv1). As Conv1 best captures early visual responses (*Figure 5—figure supplement 1*), its inclusion ensured that we did not underestimate the role of visual features in explaining neural variance. We did not use frame-wise RDMs to model these visual features; however, our approach of averaging features across video frames was justified by the short duration of our videos and the high correlation of CNN features across frames (Conv1: Pearson's $\rho = 0.89 \pm 0.09$; FC8: $\rho = 0.98 \pm 0.03$).

The analysis was carried out using 10 ms sliding windows with an overlap of 6 ms. The resulting predicted Kendall's $\tau_A$ values were tested against chance using one-tailed sign permutation testing (5000 iterations, cluster-corrected for multiple comparisons using the maximum cluster sum across time windows and regressions performed, $\alpha = 0.05$, cluster-setting $\alpha = 0.05$). The distributions of effect onsets across the 100 split-half iterations were compared using two-sided Wilcoxon signed-rank tests.

## Data availability

Behavioral and EEG data and results have been archived as an Open Science Framework repository (https://osf.io/hrmxn/). Analysis code is available on GitHub (https://github.com/dianadima/mot_action; *Dima, 2021*).

## Acknowledgements

This material is based upon work supported by the Center for Brains, Minds and Machines (CBMM), funded by NSF STC award CCF-1231216. The authors wish to thank Tara Ghazi, Seah Chang, Alyssandra Valenzuela, Melody Lee, Cora Mentor Roy, Haemy Lee Masson, and Lucy Chang for their help with the EEG data collection, Dimitrios Pantazis for pairwise decoding code, and Emalie McMahon for comments on the manuscript.

## Additional information

### Funding

| Funder | Grant reference number | Author |
| --- | --- | --- |
| National Science Foundation | CCF-1231216 | Leyla Isik |

The funders had no role in study design, data collection and interpretation, or the decision to submit the work for publication.

### Author contributions

Diana C Dima, Conceptualization, Data curation, Formal analysis, Investigation, Methodology, Project administration, Resources, Software, Validation, Visualization, Writing – original draft, Writing – review

and editing; Tyler M Tomita, Conceptualization, Data curation, Methodology, Writing – review and editing; Christopher J Honey, Conceptualization, Methodology, Writing – review and editing; Leyla Isik, Conceptualization, Funding acquisition, Methodology, Resources, Supervision, Writing – review and editing

**Author ORCIDs**
Diana C Dima http://orcid.org/0000-0002-9612-5574
Christopher J Honey http://orcid.org/0000-0002-0745-5089

**Ethics**
Human subjects: All procedures for data collection were approved by the Johns Hopkins University Institutional Review Board, with protocol numbers HIRB00009730 for the behavioral experiments and HIRB00009835 for the EEG experiment. Informed consent was obtained from all participants.

**Decision letter and Author response**
Decision letter https://doi.org/10.7554/eLife.75027.sa1
Author response https://doi.org/10.7554/eLife.75027.sa2

## Additional files

**Supplementary files**
• Supplementary file 1. Additional information about the stimulus sets and features used in analysis. (a) Breakdown of scene setting and number of agents across the two final stimulus sets. (b) Features quantified in both stimulus sets and used to generate feature representational dissimilarity matrices (RDMs) in the representational similarity analysis.

• Transparent reporting form

**Data availability**
Behavioral and EEG data and results have been archived as an Open Science Framework repository (https://osf.io/hrmxn/). Analysis code is available on GitHub (https://github.com/dianadima/mot_action, copy archived at swh:1:rev:af9eede56f27215ca38ddd32564017f1f90417d0).

The following dataset was generated:

| Author(s) | Year | Dataset title | Dataset URL | Database and Identifier |
|---|---|---|---|---|
| Dima DC, Tomita TM, Honey CJ, Isik L | 2021 | Social-affective features drive human representations of observed actions | https://osf.io/hrmxn/ | Open Science Framework, hrmxn |

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
