## [Editor Report]

This study investigates and characterizes the representations of visual actions in video stimuli. The combination of the analytical techniques and stimulus domain makes the article likely to be of broad interest to scientists interested in action representation amidst complex sequences. This article enhances our understanding of visual action representation and the extraction of such information in natural settings.

---

## [Decision Letter]

**Decision letter after peer review:**

Thank you for submitting your article "Social-affective features drive human representations of observed actions" for consideration by *eLife*. Your article has been reviewed by 2 peer reviewers, and the evaluation has been overseen by a Reviewing Editor and Chris Baker as the Senior Editor. The following individuals involved in review of your submission have agreed to reveal their identity: Angelika Lingnau (Reviewer #1); Mark Lescroart (Reviewer #2).

Essential revisions:

1) Reviewer 1 (Angelika Lingnau) notes and I agree that there are some needed qualifications/clarifications needed about the strength of the distinctions between visual, action, and social features. Given the correlations between these features, some steps need to be taken analytically to strengthen the dissociation or the claims should be attenuated. Note also her thorough smaller comments that should all be directly addressed to strengthen the paper.

2) Relatedly, there are a number of clarifications requested by Reviewer 2 (Mark Lescroart) about the construction of the features and the procedure used in the information analyses. These concerns should be carefully addressed as they will strengthen the paper.

*Reviewer #1 (Recommendations for the authors):*

Below please find a number of comments that might help strengthen the science and the presentation of this manuscript.

(1) Assignment of features to domains (e.g. page 4, line 92ff): As mentioned above, in the view of this reviewer some of the assignments between features to domains were less clear. As an example why is the setting in which the action takes place considered to be a visual feature (in contrast to, say, a semantic feature, such as indoors or outdoors)? Likewise, why are activations from the final layer of a pretrained neural network considered a high-level visual feature rather than a conceptual/semantic feature? Why is 'activity' (the amount of activity) in a video considered an action-related feature instead of a visual feature? Why is the number of agents considered a social-affective feature (instead of a visual feature; see also my next comment)? Regarding the latter, the authors admit that this feature might be considered either visual or social-affective.

(2) I liked the attempts to try to minimize the correlations between features, which is a known problem when working with naturalistic stimuli. However, in light of the significant correlations between some of the features (in particular between sociality and the number of agents), I am concerned about biases in the estimation of the β weights, which in turn may have impacted the results of the variance partitioning analysis. Moreover, if the features sociality and number of agents are correlated, and if we accept that the number of agents might be a visual rather than a social feature, how do we know that the results obtained for the feature sociality really are based on a social-affective feature?

(3) Methods (page 4, line 89): What was the reason to choose 8 additional videos with no agents?

(4) Figure 1: It might be helpful for the reader to indicate in the figure which of the features are considered to be visual, action-related or socio-affective.

(5) Figure 1B: Please provide some more details on how these plots were generated, and what they show.

(6) Multi-arrangement task (page 7, line 130f): At several points in the manuscript, the authors write that they collected behavioural similarity ratings when referring to the multi-arrangement task. This wording might be confusing since the multi-arrangement task did not require an explicit rating (in contrast to the ratings the authors collected from a separate set of participants for the features sociality, valence, arousal, activity, and transitivity). The authors may want to use a different wording when referring to the results of the multi-arrangement task.

(7) Related to the previous point, I'm somewhat concerned regarding the fact that participants were not provided with specific instructions as to the criteria they should follow when performing the multi-arrangement task. Whereas the authors argue that this was done to emphasize natural behaviour, in the view of this reviewer this approach comes with the risk of not knowing what the participants did in the end. To give concrete examples, it is possible that some participants focused on the context of the actions, while other participants focused on the valence, while yet another set of participants focused on some low level visual features. Moreover, participants might have changed their strategies throughout the experiment if they figured out that a criterion they first considered useful no longer worked for some of the configurations. Whereas the reported leave-one-subject-out correlations were above chance in both experiments, they were overall quite low, which might be related to the issue described above. How can the authors rule out such behaviour (and if not, how could this impact the interpretation of the results)?

(8) Figure 4: The overall differences between the three feature types appear to be quite small. What is the contribution of the number of agents for the social features (see also my corresponding comment on Figure 1a/ correlations between these two features)?

(9) On page 10, 2nd paragraph, the authors report the variance explained by the ten predictors in Experiment 2. What about the corresponding value for Experiment 1?

(10) Task EEG experiment (e.g. page 11, line 212f): participants had to perform a one-back action task in which they detected repetitions of the action category. Did they know what the action categories were (see also related question on the contents of Table 1)?

(11) Figure 5B: maybe add 'noise ceiling' to the legend?

(12) Discussion: the authors write that sociality did not correlate with the EEG patterns. Could it be that different participants used different strategies when performing this rating? (see corresponding comment on instructions for rating of sociality, page 22).

(12) Discussion: page 18: The authors argue for a hierarchy from visual to action-related to socially relevant computations. How might the one-back task (focusing on repetitions of the same action category) have contributed to this result?

(13): Table 1 (page 20): The mapping between the verb labels provided in Table 1 and the 8 exemplars used for each activity did not become quite clear to me. It seems from Table 1 that some of the categories consisted of a range of different actions (e.g. crying, cuddling, feeding, giggling, socializing for the activity 'childcare'), while some of them consisted of one single action only (e.g. reading), which makes the categories hard to compare, both in terms of the underlying perceptual variability and the task difficulty associated with judging a repetition of that category. Please comment.

(14) Table 1 (page 20): Unless I misunderstood, the action 'socializing' occurred within four of the 18 activities (in addition to being one of the 18 activities). What was the reason for this? Could this have impacted the results, showing a stronger contribution of socio-affective features in comparison to action-related and visual features? Likewise, could the results obtained in this study be affected by the curation of the two stimulus datasets (adjusted to include both social and non-social activities)?

(15) Page 20: What was the reason to use these specific common activities? Was there a specific cutoff?

(16) Page 21, line 404f: What was the purpose of the control videos depicting natural and indoors scenes?

(17) Page 21, line 415f: Please explain what 'repetitive' videos mean in this context.

(18) Page 22, behavioural ratings: What were the exclusion criteria for the behavioural ratings and the multiple arrangement task?

(19) Page 22: Participants were instructed to rate 'how social the events were'. In the view of this reviewer, this leaves a lot of room for interpretation (and also differs from the instructions given in previous studies). Please comment.

(20) Page 22: How consistent were the behavioural ratings across participants? Was variability across participants higher for some features than for others?

(21) Page 23 line 454f: How was the subset of videos presented in trial 1 selected?

(22) Page 28 line 541f: According to which criteria did the authors center the 0.5 seconds window around the action?

(23) Page 28 line 557f: What do the authors mean by 'shuffled pairs'?

(24) Page 29, 2nd paragraph: Please provide details on the projector.

*Reviewer #2 (Recommendations for the authors):*

If I have read the paper (and code) correctly, it seems that a single RDM was used to model all timepoints in the EEG responses, and that single RDM was correlated with the RDMs generated by the pairwise decoding analysis for each timepoint. For purposes of modeling behavioral data – in which the subjects are making a judgment based on the entire video clip – a single RDM per video clip seems reasonable. However, for purposes of modeling time-varying neural responses, it may be less than ideal to use a single RDM to summarize each video. The usefulness of a single temporal summary RDM will depend on how homogenous the features are across the video clip. Having looked over a few of the video clips in Moments in Time, the clips appear to be generally slow-moving – i.e., no large changes in background elements or actors across the 3 seconds. This seems like a reasonable justification for the use of a single temporal summary RDM; I would encourage the authors to clarify their rationale for using one RDM for the whole timecourse instead of framewise RDMs. Quantitative arguments would be useful.

[Editors’ note: further revisions were suggested prior to acceptance, as described below.]

Thank you for resubmitting your work entitled "Social-affective features drive human representations of observed actions" for further consideration by *eLife*. Your revised article has been evaluated by Chris Baker (Senior Editor) in consultation with the original reviewers.

We all agree the manuscript has been improved, the issues raised by the reviewers have been addressed well, and the manuscript will make a nice addition to the literature. But one of the reviewers highlighted a remaining concern that I wanted to draw your attention to. This concern could be addressed with additional analyses or additional discussion and motivation of the approach you used (I will leave it up to you which avenue you want to pursue). I'll quote from the reviewer:

"Among the visual features the authors investigate is motion. To me, this is a visual feature that would seem to have a strong potential to explain the EEG data in particular, since motion contrast strongly drives many visual areas. I am not completely satisfied with the way the authors have quantified motion, and I have a medium concern that their motion model is a bit of a straw man. First, they refer to the model as a "motion energy" model, which is not technically correct. Motion energy (i.e. Adelson and Bergen, 1985; Nishimoto et al., 2011) reflects image contrast, whereas optic flow does not (at least to the same degree). For example, a gray square moving 2 degrees on a slightly less gray background will generate the same magnitudes of optic flow vectors as a starkly black square on a white background moving the same distance. Motion energy will be substantially greater for higher-contrast stimuli. As such, it's likely that motion energy would be a better model for visual responses (at least in early visual areas) than optic flow. They also choose to compute optic flow densely, with one value per pixel. Thus, if actors in the various videos are in slightly different locations, the optic flow metric could be quite different, and the RDM may characterize arguably similar stimuli as distinct. I think that a multi-scale pyramid of optic flow (or better still, motion energy) would have a better chance of capturing motion selectivity in the way that the brain does.

There is one more choice that the authors make that I'm not entirely comfortable with, which relates to this same issue. They choose to use only the models that can be related to behavioral data to model the EEG data. This choice is motivated by the desire to relate brain activity to behavior, which is a worthwhile endeavor. However, I think there is an assumption underlying this choice: the authors assume that the signal that they measure with EEG must reflect neural representations that guide behavior and not low-level perceptual processes. I do not think that this is guaranteed to be the case. EEG measures a spatially diffuse signal, which may well be dominated by activity from quite low-level areas (e.g. V1). I think the question of whether the EEG signal reflects relatively high-level cognitive processing or relatively low-level perceptual processing – for any given experiment – is still open. For most of the models they exclude, I don't think this is a big issue. For example, I think it's reasonable to test luminance basically as a control (to show that there aren't huge differences in luminance that can explain behavior) and then to exclude it from the EEG modeling. However, I'm less happy with exclusion of motion, based on a somewhat sub-optimal motion model not explaining behavior.

The combination of these two issues has left me wanting a better quantification of motion as a model for the EEG data. My bet would be that motion would be a decent model for the EEG signal at an earlier stage than they see the effects of action class and social factors in the signal, so I don't necessarily think that modeling motion would be likely to eliminate the effects they do see; to my mind, it would just present a fairer picture of what's going on with the EEG signal."

---

## [Author Response]

Reviewer #1 (Recommendations for the authors):Below please find a number of comments that might help strengthen the science and the presentation of this manuscript.(1) Assignment of features to domains (e.g. page 4, line 92ff): As mentioned above, in the view of this reviewer some of the assignments between features to domains were less clear. As an example why is the setting in which the action takes place considered to be a visual feature (in contrast to, say, a semantic feature, such as indoors or outdoors)? Likewise, why are activations from the final layer of a pretrained neural network considered a high-level visual feature rather than a conceptual/semantic feature? Why is 'activity' (the amount of activity) in a video considered an action-related feature instead of a visual feature? Why is the number of agents considered a social-affective feature (instead of a visual feature; see also my next comment)? Regarding the latter, the authors admit that this feature might be considered either visual or social-affective.

This is an important point, and we agree the distinction between feature domains is not always clear cut. To answer the question about why the setting and final DNN layer are considered visual: since this work focused on actions, we considered action-related features to pertain to actions and their semantics. Thus, we considered the setting of an action to be visual and not semantic, since it only pertains to the action’s context. Similarly, the neural network we used was pretrained on ImageNet, and as such the features extracted by it would tend to describe the environment and objects involved rather than the action.

The word ‘visual’, particularly when contrasted with ‘action’ and ‘social-affective’, may indeed suggest only low-level information; however, we believe the distinction between context, actions, and agents to be relevant as it captures the variability of naturalistic actions. For example, an action like ‘eating’ would be characterized by action features (action category, transitivity, effectors involved and amount of activity) that would stay largely consistent across exemplars. However, exemplars may vary in terms of context (eating in the kitchen vs at a park), object (eating an apple vs a sandwich), and agents (eating alone vs together). We clarified these points in the paper, when introducing the visual features :

“Naturalistic videos of actions can vary along numerous axes, including visual features (e.g. the setting in which the action takes place or objects in the scene), action-specific features (e.g. semantic action category), and social-affective features (e.g. the number of agents involved or perceived arousal). For example, an action like ‘eating’ may vary in terms of context (in the kitchen vs at a park), object (eating an apple vs a sandwich), and number of agents (eating alone vs together). Drawing these distinctions is crucial to disambiguate between context, actions, and agents in natural events. […] Visual features ranged from low-level (e.g. pixel values) to high-level features related to scenes and objects (e.g. activations from the final layer of a pretrained neural network).”

Combining these scene-related features with action-related features does, indeed, reach an explanatory power almost as high as that of social-affective features in Experiment 1 (but not in Experiment 2). This can also be inferred from our original Figure 4 by looking at the effect sizes.

In our view, this strengthens our conclusions. Considering that this analysis discounts any variance shared with high-level visual, semantic, and action features, the significantly greater unique contribution of social-affective features is striking. However, the difference in effect size between the two experiments becomes more apparent than in our previous analysis (Figure 4), and we now discuss this in the manuscript :

“Although the assignment of features to domains was not always straightforward, our results were robust to alternative assignment schemes. For example, high-level visual features can be seen as bordering the semantic domain, while features like the number of agents or the amount of activity can be seen as visual. However, feature assignment was not the main factor driving our results, which stayed the same even when the activity feature was assigned to the visual group. More strikingly, the social-affective feature group explained significantly more variance than all other features grouped together in both experiments (Figure 4 —figure supplement 2). This is a particularly stringent test as it pits the unique and shared contributions of all visual, semantic, and action features against the four social-affective features. In Experiment 1, the combined contribution of visual and action features approached that of social-affective features, while in Experiment 2 the difference was larger. Together with the larger contribution of the number of agents in Experiment 2, this suggests that Experiment 2 may have captured more social information, potentially thanks to the exhaustive sampling of the stimuli which allowed each participant to arrange the videos according to different criteria.”

We also agree that the number of agents straddles the boundary between visual and social features. However, our control analysis showed that the number of agents did not explain the unique contribution of social-affective features to behavior (Figure 4 —figure supplement 3 ).

Finally, we took “activity” to pertain to actions (is it a static action like reading, or a high-movement action like running?) We agree that it could also be considered a visual feature along the lines of motion energy. However, the assignment of this feature to the visual group does not impact our conclusions, which we now note :

“However, feature assignment was not the main factor driving our results, which stayed the same even when the activity feature was assigned to the visual group.”

**Author response image 1. sa2fig1:** Unique contributions of visual features (environment, FC8, activity), action features (action category, effectors, transitivity) and social-affective features (number of agents, sociality, valence and arousal).

**Author response image 2. sa2fig2:** Social-affective features (sociality, valence, and arousal) still contribute more variance than visual features (environment, FC8, activity, and number of agents) and action features (action category, transitivity, and effectors).

(2) I liked the attempts to try to minimize the correlations between features, which is a known problem when working with naturalistic stimuli. However, in light of the significant correlations between some of the features (in particular between sociality and the number of agents), I am concerned about biases in the estimation of the β weights, which in turn may have impacted the results of the variance partitioning analysis. Moreover, if the features sociality and number of agents are correlated, and if we accept that the number of agents might be a visual rather than a social feature, how do we know that the results obtained for the feature sociality really are based on a social-affective feature?

Indeed, despite our efforts, there were still moderate correlations between certain related features. However, the variance inflation factors were low (1.34 in Exp 1 and 1.37 in Experiment 2), suggesting that the regressions did not suffer from collinearity. Furthermore, different assignments of features to groups (see above) did not impact the overall pattern of results.

It is true that these analyses do not allow us to draw conclusions about individual features, as variance partitioning with more than three sets of features becomes intractable. Specifically, the role of sociality and number of agents is unclear.

We performed an additional analysis looking at the unique contributions of the number of agents, sociality, and the grouped affective features (valence and arousal): Figure 4 —figure supplement 4. This has the advantage of removing any shared variance between the number of agents and sociality. Based on this analysis, it appears that valence and arousal best accounted for behavioral similarity estimates, with the number of agents also contributing a significant amount, particularly in Experiment 2. However, this analysis does not partial out the contributions of other features (visual or action-related). Grouping the two affective features together may have also advantaged them over the single social features. Given these limitations and the post-hoc, exploratory nature of this new analysis, we prefer to focus our interpretation on feature domains and their contributions. Nonetheless, we have adjusted our claims in light of this new finding, as outlined below.

We discuss this additional analysis in the manuscript :

“Furthermore, an additional analysis looking at the separate contributions of the number of agents, sociality, and affective features (valence and arousal) found that the affective features contributed the greatest variance in both experiments (Figure 4 —figure supplement 4). For technical reasons, this analysis compared the joint contribution of both affective features to each single social feature and did not discount the impact of variance shared with visual or action-related features. Despite these limitations, the results suggest that the contribution of the social-affective feature group is not driven by the number of agents or the variance it shares with sociality, and highlight the role of affective features (valence and arousal) in explaining behavior.”

We also discuss these findings in the Discussion :

“An exploratory follow-up analysis showed that this effect was primarily driven by affective features (valence and arousal), with the number of agents as a secondary contributor. Recent work found that affective features drive the perceived similarity of memories of real-life events (Tomita et al., 2021), suggesting that these features bridge the action, event and memory domains in organizing mental representations.”

Interestingly, this new finding is in line with our EEG results, where the number of agents, valence and arousal correlated with the neural responses, while sociality did not. Although this is an exploratory finding, we briefly discuss it :

“Note that this finding is mirrored in our behavioral results, where we observed larger unique contributions from valence, arousal and the number of agents than sociality (Figure 4 —figure supplement 4).”

(3) Methods (page 4, line 89): What was the reason to choose 8 additional videos with no agents?

We added this explanation to page 4 , as well as the Methods :

“and 8 additional videos with no agents, included to add variation in the dataset, see Materials and methods, section (*Behavior: Stimuli*)”

“The videos with no agents were included to provide variation in the dataset along visual properties that did not pertain to actions or agents, as well as variation in the overall dataset in terms of number of agents per video.”

(4) Figure 1: It might be helpful for the reader to indicate in the figure which of the features are considered to be visual, action-related or socio-affective.

We added a color-coded legend to the figure to make it clearer at a glance how features were assigned to groups .

(5) Figure 1B: Please provide some more details on how these plots were generated, and what they show.

We added the following caption to this figure panel :

“Behavioral rating distributions in the two stimulus sets. The z-scored ratings were visualized as raincloud plots showing the individual data points, as well as probability density estimates computed using Matlab’s *ksdensity* function (Allen et al., 2019).”

This panel shows the distribution of labeled features across our two stimulus sets, in particular that the ratings were consistent and stimuli spanned a wide range along each feature.

(6) Multi-arrangement task (page 7, line 130f): At several points in the manuscript, the authors write that they collected behavioural similarity ratings when referring to the multi-arrangement task. This wording might be confusing since the multi-arrangement task did not require an explicit rating (in contrast to the ratings the authors collected from a separate set of participants for the features sociality, valence, arousal, activity, and transitivity). The authors may want to use a different wording when referring to the results of the multi-arrangement task.

Thank you for bringing this to our attention. Indeed, these were not ratings in the sense that the other features were behaviorally rated. We replaced such wording throughout the manuscript with either ‘similarity estimates’ or ‘similarity judgments’. (This latter wording is more accurate in our view, since participants were instructed to judge the similarity of the videos in order to place them on screen.) See pages 7, 8, 27, 31, 43 etc.

(7) Related to the previous point, I'm somewhat concerned regarding the fact that participants were not provided with specific instructions as to the criteria they should follow when performing the multi-arrangement task. Whereas the authors argue that this was done to emphasize natural behaviour, in the view of this reviewer this approach comes with the risk of not knowing what the participants did in the end. To give concrete examples, it is possible that some participants focused on the context of the actions, while other participants focused on the valence, while yet another set of participants focused on some low level visual features. Moreover, participants might have changed their strategies throughout the experiment if they figured out that a criterion they first considered useful no longer worked for some of the configurations. Whereas the reported leave-one-subject-out correlations were above chance in both experiments, they were overall quite low, which might be related to the issue described above. How can the authors rule out such behaviour (and if not, how could this impact the interpretation of the results)?

In this work, we specifically focused on intuitive (and thus unconstrained) similarity. Indeed, this means that participants could have used different criteria or changed their strategies over time; however, we argue that this is a strength of our approach. We used the iterative multiple arrangement task as implemented in Meadows specifically because by varying the groups of stimuli shown in each trial it is able to recover a multidimensional representation, with many potential features coming into play. Although this method is not as exhaustive as, for example, an odd-one-out task (e.g. Hebart et al., 2021), it was a good compromise in terms of handling a large stimulus set and recovering a multidimensional structure. While the leave-one-subject-out correlations were only moderate, they reached a similar level as seen in related prior studies (e.g., Tarhan et al., Neuropsychologia, 2021) with smaller, more homogenous datasets. Finally, we note that all reported measures in the paper are in terms of Kendall’s Tau-A, which is a more conservative metric than the more commonly reported Spearman’s correlation.

We also argue that our ability to find and replicate the main organizing axes of this space, despite the many sources of potential variability you outline above, suggests that there is a common element to participants’ multiple arrangement strategies. Recent work on intuitive action understanding supports this by suggesting that intuitive arrangements may be guided by judgments of action goals (Tarhan et al., Neuropsychologia, 2021). We have added the following lines to the Results :

“The multiple arrangement task was unconstrained, which meant that participants could use different criteria. Although this may have introduced some variability, the adaptive algorithm used in the multiple arrangement task enabled us to capture a multidimensional representation of how actions are intuitively organized in the mind, while at the same time ensuring sufficient data quality.”

(8) Figure 4: The overall differences between the three feature types appear to be quite small. What is the contribution of the number of agents for the social features (see also my corresponding comment on Figure 1a/ correlations between these two features)?

Although the differences are small, they are significant relative to each other and as a percentage of the true correlation. In Experiment 1, social-affective features explain more than twice the unique variance explained by visual and action features (although note that the lower reliability of this data makes the true correlation more difficult to trust as a measure of ‘best possible fit’). In Experiment 2, social-affective features uniquely contribute 58% of the true correlation, with only 9% contributed by action features and approximately 0.5% contributed by visual features. We added more information on this to the Results:

“Although the effect sizes were relatively low, social-affective features explained more than twice as much unique variance as either the visual or action features in Experiment 1, and six times as much in Experiment 2. Furthermore, given the limits placed on predictivity by the reliability of the behavioral data, affective features predicted a large portion of the explainable variance in both experiments.”

We had not analyzed the unique contributions of each feature within each group, as this is difficult to assess in the context of a variance partitioning analysis given the large number of features (please see our response to point 2). Our above analyses (see Figure 4 – supplements 3 and 4) suggest that the social-affective dominance is not driven by the number of agents.

(9) On page 10, 2nd paragraph, the authors report the variance explained by the ten predictors in Experiment 2. What about the corresponding value for Experiment 1?

We have edited the text to make this clearer. The section now reads as follows :

“In Experiment 1, the predicted squared Kendall’s τ_A_ of the full model (τ_A_^2^=0.06±0.001 ) was higher on average than the true split-half squared correlation (τ_A_^2^=0.04±0.002). This is likely to be due to the lower reliability of the behavioral similarity data in this experiment, and suggests that the ten predictors are able to explain the data well despite the overall lower prediction accuracy.”

Due to the lower reliability of this dataset , we did not want to over-interpret the prediction accuracy obtained here.

(10) Task EEG experiment (e.g. page 11, line 212f): participants had to perform a one-back action task in which they detected repetitions of the action category. Did they know what the action categories were (see also related question on the contents of Table 1)?

The task was not constrained by the action categories used in our dataset curation. Participants were simply instructed to press a button when they identified the same action in consecutive videos, and given ‘eating’ as a simple example. However, the catch video pairs contained more easily identifiable repeated actions (e.g. people doing ballet rather than people doing different sports, which might have been more difficult to label as a repeat). This ensured that participants could perform the task without biasing their semantic categorization. We have clarified this in the text :

“Participants were not instructed on what constituted an action, beyond being given “eating” as a simple example.”

(11) Figure 5B: maybe add 'noise ceiling' to the legend?

Thank you. We have added this .

(12) Discussion: the authors write that sociality did not correlate with the EEG patterns. Could it be that different participants used different strategies when performing this rating? (see corresponding comment on instructions for rating of sociality, page 22).

This is an interesting point. We assessed the reliability of the ratings using leave-one-subject-out correlations, and did not find sociality to be less consistent than other ratings (Figure 1 —figure supplement 1). In Experiment 1, sociality was more reliable than all other ratings, and significantly more reliable than valence, arousal and activity ratings. In Experiment 2, there was no significant difference between sociality and the other ratings, except transitivity, which was more reliable. The fact that we do see EEG correlations with other features like valence and arousal seems to suggest that the lack of an effect is not due to differences in reliability. We now mention this point :

“This effect was not likely to be driven by a lower reliability in the measurement of this feature, as sociality was more reliable than all other behaviorally rated features in Experiment 1 (Figure 1 —figure supplement 1).”

We also now discuss the reliability of the behavioral ratings (see also below comment 20).

(12) Discussion: page 18: The authors argue for a hierarchy from visual to action-related to socially relevant computations. How might the one-back task (focusing on repetitions of the same action category) have contributed to this result?

We thank the Reviewer for this interesting point. Although the task was not related to most of the features, it required a degree of explicit, semantic processing that may have contributed to the neural dynamics. Although we cannot fully answer this question with the current data, we added the following paragraph to the discussion :

“Importantly, these features emerged spontaneously, as the one-back task performed during the EEG recordings only related to action category. However, the semantic processing required to perform the task may have contributed to these computations. The emergence of features irrelevant to the task at hand (action category is not correlated with any other features in our dataset) suggests that this temporal hierarchy would also emerge in the absence of a task; however, future work can more directly test the impact of implicit and explicit (e.g. social-affective) processing on these neural dynamics.”

(13): Table 1 (page 20): The mapping between the verb labels provided in Table 1 and the 8 exemplars used for each activity did not become quite clear to me. It seems from Table 1 that some of the categories consisted of a range of different actions (e.g. crying, cuddling, feeding, giggling, socializing for the activity 'childcare'), while some of them consisted of one single action only (e.g. reading), which makes the categories hard to compare, both in terms of the underlying perceptual variability and the task difficulty associated with judging a repetition of that category. Please comment.

The reviewer raises an important point. The reason for this partially stems from the imperfect mapping between our broader, ATUS-based categories and the verb labels in the Moments in Time (MiT) dataset. This is related to broader issues with category labels in computer vision datasets, which are often criticized for not being meaningful distinctions for humans (e.g., the dozens of different dog breeds in Imagenet). We sought to divide the MiT videos into more meaningful categories as defined by ATUS, but due to the somewhat arbitrary nature of the verb labels in MiT, there is not a perfect one-to-one mapping. Some of the categories mapped directly onto the labels, while other categories were broader and represented by several labels. Furthermore, many videos could have been represented by several other labels in addition to the ones they were listed under (hence why some ‘driving’ videos were found under ‘socializing’, for example).

Our original curation procedure involved going through the MiT verb labels and selecting videos from any label that seemed relevant for each ATUS activity category. This led to an initial selection of approximately 500 videos, from which the two datasets were selected solely based on the ATUS activity categories without regard to the MiT verb labels (as described in the Methods). Although the verb labels may seem heterogeneous, we attempted to enhance and match perceptual variability across categories by selecting videos for each ATUS category that varied along different criteria such as environment and number of agents. For example, the ‘reading’ category, though represented by a single label in the Moments in Time dataset, contains videos depicting one or two agents of different ages, genders and ethnicities reading books or magazines both indoors (in a living room, in bed, in a library, on the subway etc.) and outdoors (in a park, by a river, in a hammock etc.). We have added this to the manuscript :

“Videos were chosen that were horizontally-oriented (landscape), of reasonable image quality, that clearly represented the activity in question, that clearly depicted an agent performing the activity, and that varied in terms of number of agents, gender and ethnicity of agents, and scene setting. Although some categories were represented by fewer verb labels than others in the final set, our curation procedure aimed to balance important features within and across action categories.”

Finally, any remaining heterogeneity across action categories was addressed through our feature-driven analysis approach, which was intended to capture any perceptual components that may affect how participants arrange the videos.

(14) Table 1 (page 20): Unless I misunderstood, the action 'socializing' occurred within four of the 18 activities (in addition to being one of the 18 activities). What was the reason for this? Could this have impacted the results, showing a stronger contribution of socio-affective features in comparison to action-related and visual features? Likewise, could the results obtained in this study be affected by the curation of the two stimulus datasets (adjusted to include both social and non-social activities)?

Our aim was to sample actions that involved both single and multiple agents in a balanced way, as this has been rarely done in previous work using controlled videos of actions, usually performed by one agent. Thus, some of the actions were found under the label ‘socializing’ and involved multiple agents performing the action (whether interacting or not). This is also due to the way the Moments in Time labels were assigned, as described above. The presence of a socializing category, however, was driven by the prominence assigned to this category in the 2019 American Time Use Survey under the broader ‘leisure’ umbrella. We acknowledge that the distinctions from ATUS are not perfect, but provided the best objective measure to curate our dataset. We have added a brief description of these shortcomings .

“We note that the ATUS distinctions are based on performed rather than observed actions. While imperfect, they provide an ecologically-relevant and objective criteria with which to define our action space.”

To your main concern, we do not see the presence of both social and non-social actions as a source of bias. First, most action categories were represented by both single-agent and multiple-agent videos. Although some categories inherently involved multiple people (socializing, instructing), others tended to involve single agents and no social aspect (sleeping, housework). Second, all features (including sociality and number of agents) were rated independently of action category, and feature distributions suggest that the stimuli varied along many axes (Figure 1b). Finally, each feature’s contribution to behavior was assessed independently of category labels.

Ultimately, we believe that an action space that includes both social and non-social activities is more representative of human actions in the real world, since many actions are directed towards others. On the other hand, no stimulus set is definitive, and it would be great to replicate these results with different types of stimuli. The question of how action categories should be defined is still an open one that we did not attempt to address here – although we argue that our feature-based approach mitigates this somewhat.

We have clarified our reasoning in text :

“Our final dataset included 18 social and non-social activities that lend themselves to visual representation (Table 1), to ensure a diverse and balanced stimulus set representative of the human everyday action space.”

(15) Page 20: What was the reason to use these specific common activities? Was there a specific cutoff?

We have clarified our selection of ATUS categories in the manuscript . The selected activities were second-level ATUS activities with a cutoff of at least 0.14 hours/day, with few exceptions (additions and omissions) that are now explicitly described in text :

“Action categories were selected from the second-level activities identified in the ATUS. We used a minimum cutoff of 0.14 hours/day to select common actions (Table 1). To diversify our dataset, we added a ‘hiking’ category (to increase variability in scene setting) and a “fighting” category (for variability along affective dimensions). In addition, “driving” was selected as a more specific instance of the “travel” categories in ATUS, as it is the most common form of transportation in the US. Some adjustments were also made to the “relaxing and leisure” category, by selecting two specific activities that were easy to represent and distinguish visually, as well as above threshold (“reading” and “playing games”). In addition, our “reading” category included both “reading for personal interest” and “homework and research”, as this distinction is difficult to convey visually. We omitted three leisure categories that were difficult to represent in brief videos (“watching TV”, “relaxing and thinking”, and “computer use for leisure, excluding games”), as well as the ”consumer goods purchases” category.”

We have also added the number of hours per day spent on each activity according to ATUS to Table 1.

(16) Page 21, line 404f: What was the purpose of the control videos depicting natural and indoors scenes?

We added this explanation to the Methods. Please see response to point 3 above.

(17) Page 21, line 415f: Please explain what 'repetitive' videos mean in this context.

After running our randomization procedure, there were a few videos within the same action category that shared a high degree of visual similarity (e.g. involving a similar number of agents in similar postures). The issue of visually similar stimuli was not apparent in the larger Experiment 1 set. We clarified our wording in text:

“From the remaining videos, a second set of 76 videos was sampled and manually adjusted to remove videos without agents (in the interest of experimental time) and any videos that were too similar visually to other videos in the same category (e.g., involving a similar number of agents in similar postures).”

(18) Page 22, behavioural ratings: What were the exclusion criteria for the behavioural ratings and the multiple arrangement task?

We thank the reviewer for spotting this oversight. We added this information in text:

“Participants were excluded if they responded incorrectly to catch trials (approximately 10% of trials) requiring them to label the action shown in the prior video, or if they provided overly repetitive ratings (e.g. using only two unique values or fewer out of five possible ratings throughout the experiment).”

(19) Page 22: Participants were instructed to rate 'how social the events were'. In the view of this reviewer, this leaves a lot of room for interpretation (and also differs from the instructions given in previous studies). Please comment.

We agree that our instructions did not much constrain participants’ judgments of sociality. Given the diversity of our natural videos (which involved interactions as well as people simply acting alongside each other), we wanted to capture the full spectrum of what sociality means to people, rather than focusing on a yes/no detection of social interaction. Previous studies are also quite heterogeneous in their measurement of sociality, with some focusing on interactions (e.g. Tucciarelli et al., 2019) and others defining it as person-directedness (Tarhan and Konkle, 2020) or relevance of one agents’ actions to another (Wurm and Caramazza 2019, Wurm, Caramazza and Lingnau 2017). In our view, the validity of our instructions is supported by the fact that sociality ratings were not less reliable than other ratings and among the most reliable of our social ratings. On the other hand, the different operationalization means that this feature may have captured different information than in previous studies, and may explain our lack of correlation between sociality and EEG data. We now mention this in the discussion:

“Alternatively, our operationalization of this feature (which was broader than in some previous studies, e.g. Tucciarelli et al., 2019; Wurm et al., 2017) may have led to differences in the information captured.”

(20) Page 22: How consistent were the behavioural ratings across participants? Was variability across participants higher for some features than for others?

This information is summarized in Figure 3 —figure supplement 1 and its caption. We now also discuss this in text:

“Behaviorally rated features differed in reliability in Experiment 1 (F(4,819) = 22.35, P<0.001), with sociality being the most reliable and arousal the least reliable (Figure 3 —figure supplement 1). In Experiment 2, however, there was no difference in reliability (F(4,619) = 0.76, P=0.55). Differences in reliability were mitigated by our use of feature averages to generate feature RDMs.”

(21) Page 23 line 454f: How was the subset of videos presented in trial 1 selected?

We added some more information on this procedure in text:

“The experiments included a training trial in which participants arranged the same seven videos (in Experiment 1) or eight videos (in Experiment 2) before beginning the main task. In both experiments, these videos were hand selected to represent clear examples from four categories.”

(22) Page 28 line 541f: According to which criteria did the authors center the 0.5 seconds window around the action?

The procedure was based on visual inspection of the videos. In most cases, the first 0.5 seconds were selected; however, a different segment was selected in cases where this made the action clearer (e.g. a video that began with someone catching a ball to then throw it was cropped around the throwing action, which could then be shown in its entirety). This was especially important given the brief nature of the 0.5 s videos. We have clarified this in text:

“The three-second stimuli were trimmed to a duration of 0.5 seconds centered around the action as determined by visual inspection, to ensure that the shorter videos were easily understandable. This helped improve time-locking to the EEG signals and allowed for a condition-rich experimental design.”

(23) Page 28 line 557f: What do the authors mean by ‘shuffled pairs’?

We clarified what we meant:

“The video pairs were presented in different orders to minimize learning effects, so that for each video pair (V1, V2), half of the presentations were in the order V1-V2 and half of them were in the order V2-V1.”

(24) Page 29, 2^nd^ paragraph: Please provide details on the projector.

We added this information:

“The stimuli were presented using an Epson PowerLite Home Cinema 3000 projector with a 60Hz refresh rate.”

Reviewer #2 (Recommendations for the authors):If I have read the paper (and code) correctly, it seems that a single RDM was used to model all timepoints in the EEG responses, and that single RDM was correlated with the RDMs generated by the pairwise decoding analysis for each timepoint. For purposes of modelling behavioral data – in which the subjects are making a judgment based on the entire video clip – a single RDM per video clip seems reasonable. However, for purposes of modelling time-varying neural responses, it may be less than ideal to use a single RDM to summarize each video. The usefulness of a single temporal summary RDM will depend on how homogenous the features are across the video clip. Having looked over a few of the video clips in Moments in Time, the clips appear to be generally slow-moving – i.e., no large changes in background elements or actors across the 3 seconds. This seems like a reasonable justification for the use of a single temporal summary RDM; I would encourage the authors to clarify their rationale for using one RDM for the whole timecourse instead of framewise RDMs. Quantitative arguments would be useful.

We agree that a time-resolved approach to modelling the video features could offer more information. Here, however, the videos used in the EEG paradigm were very brief – only 0.5 seconds long. Furthermore, we made efforts to ensure we selected videos where actions were clear and continuous, without sudden camera or actor changes. The shorter 0.5s duration of EEG stimuli likely increased visual homogeneity throughout each clip.

Finally, this would only apply to the visual features (specifically, the layer activations from AlexNet), as the other features we used for our EEG analysis did not vary across the frames (environment, number of agents, and behavioral judgments based on the entire clips). Still, to ensure that we maximized the amount of information present in our RDMs, our procedure for generating AlexNet features entailed extracting these features from each frame and averaging them across the 12 frames that the 0.5 s clips consisted of. Importantly, we did not see significant changes in AlexNet features across these frames.

In Author response image 3, we plotted the average correlation across frames between Conv1 and FC8 features extracted from AlexNet. For Conv1, the correlation was on average r = 0.89, SD = 0.09 ; for FC8, it was r = 0.98, SD = 0.03. Although there is more variability in correlations for Conv1, they are still high enough to justify our decision to select the average layer activation across frames. Furthermore, this variability diminishes in the FC8 features, which are expected to capture more high-level information, thus supporting our idea that the videos are sufficiently consistent over time to justify our approach:

**Author response image 3. sa2fig3:** Average correlation of CNN features across frames. Each dot in the scatterplots is a video, with the distribution of the correlations shown above.

We added the following to the manuscript:

“We did not use frame-wise RDMs to model these visual features; however, our approach of averaging features across video frames was justified by the short duration of our videos and the high correlation of CNN features across frames (Conv1: Pearson’s ρ= 0.89±0.09; FC8: ρ= 0.98±0.03).”

[Editors’ note: further revisions were suggested prior to acceptance, as described below.]

We all agree the manuscript has been improved, the issues raised by the reviewers have been addressed well, and the manuscript will make a nice addition to the literature. But one of the reviewers highlighted a remaining concern that I wanted to draw your attention to. This concern could be addressed with additional analyses or additional discussion and motivation of the approach you used (I will leave it up to you which avenue you want to pursue). I’ll quote from the reviewer:“Among the visual features the authors investigate is motion. To me, this is a visual feature that would seem to have a strong potential to explain the EEG data in particular, since motion contrast strongly drives many visual areas. I am not completely satisfied with the way the authors have quantified motion, and I have a medium concern that their motion model is a bit of a straw man. First, they refer to the model as a “motion energy” model, which is not technically correct. Motion energy (i.e. Adelson and Bergen, 1985; Nishimoto et al., 2011) reflects image contrast, whereas optic flow does not (at least to the same degree). For example, a gray square moving 2 degrees on a slightly less gray background will generate the same magnitudes of optic flow vectors as a starkly black square on a white background moving the same distance. Motion energy will be substantially greater for higher-contrast stimuli. As such, it's likely that motion energy would be a better model for visual responses (at least in early visual areas) than optic flow. They also choose to compute optic flow densely, with one value per pixel. Thus, if actors in the various videos are in slightly different locations, the optic flow metric could be quite different, and the RDM may characterize arguably similar stimuli as distinct. I think that a multi-scale pyramid of optic flow (or better still, motion energy) would have a better chance of capturing motion selectivity in the way that the brain does.

We have removed our reference to the optic flow model as a motion energy model, and added a new motion energy model:

“To assess whether whether a motion energy model (Adelson and Bergen, 1985; Nishimoto et al., 2011; Watson and Ahumada, 1985) would better capture the impact of motion, we performed control analyses by computing motion energy features for each video using a pyramid of spatio-temporal Gabor filters with the *pymoten* package (Nunez-Elizalde et al., 2021).”

As the Reviewer predicted, the motion energy model correlated better with behavior than our optic flow model. This correlation was significant in Experiment 1, but not in Experiment 2, in line with our previous results regarding the role of visual features across the two experiments. In the variance partitioning analysis, the addition of the motion energy model to the visual feature group did not change the pattern of results in either experiment. We describe these findings in text:

“Similarly, our decision to quantify motion and image properties separately by using an optic flow model may have reduced the explanatory power of motion features in our data. Indeed, a motion energy model (Adelson and Bergen, 1985; Nunez-Elizalde et al., 2021) significantly correlated with behavior in Experiment 1, but not in Experiment 2. However, the addition of this model did not change the pattern of unique feature contributions (Figure 4 —figure supplement 2).”

There is one more choice that the authors make that I'm not entirely comfortable with, which relates to this same issue. They choose to use only the models that can be related to behavioral data to model the EEG data. This choice is motivated by the desire to relate brain activity to behavior, which is a worthwhile endeavor. However, I think there is an assumption underlying this choice: the authors assume that the signal that they measure with EEG must reflect neural representations that guide behavior and not low-level perceptual processes. I do not think that this is guaranteed to be the case. EEG measures a spatially diffuse signal, which may well be dominated by activity from quite low-level areas (e.g. V1). I think the question of whether the EEG signal reflects relatively high-level cognitive processing or relatively low-level perceptual processing – for any given experiment – is still open. For most of the models they exclude, I don't think this is a big issue. For example, I think it's reasonable to test luminance basically as a control (to show that there aren't huge differences in luminance that can explain behavior) and then to exclude it from the EEG modeling. However, I'm less happy with exclusion of motion, based on a somewhat sub-optimal motion model not explaining behavior.The combination of these two issues has left me wanting a better quantification of motion as a model for the EEG data. My bet would be that motion would be a decent model for the EEG signal at an earlier stage than they see the effects of action class and social factors in the signal, so I don't necessarily think that modeling motion would be likely to eliminate the effects they do see; to my mind, it would just present a fairer picture of what's going on with the EEG signal."

In selecting only models that explained behavior for our EEG analysis, we did not intend to assume that EEG activity only reflects behaviorally relevant features. By adding Conv1 to the behaviorally relevant visual models (FC8 and environment), we hoped we would capture low-level visual processing while at the same time keeping the number of features tractable and linking brain and behavior (as noted by the Reviewer). However, we agree that this approach neglects the role of motion, and we have performed new analyses with the motion energy model.

Indeed, the motion energy model correlated with our EEG data during a sustained time window, with a peak during early time windows. The addition of motion energy to the group of visual features in the variance partitioning analysis increased the unique contribution of visual features and decreased the contribution of action features, suggesting that the ‘action’ portion of the signal we originally detected included some shared variance with motion. However, the temporal hierarchy was not changed in the fixed-effects analysis, which leads us to believe that the three feature groups explain distinct portions of the signal. Most importantly, the portion of variance uniquely explained by social-affective features was unchanged by the addition of the motion energy model.

We describe these results in text:

“Motion has been shown to drive the response of visual areas to naturalistic stimuli (Russ and Leopold, 2015). To better assess the effect of motion on EEG responses, we performed an additional analysis including the motion energy model. There was a sustained correlation between motion energy and EEG patterns beginning at 62 ms (Figure 6 —figure supplement 3). In the variance partitioning analysis, the addition of motion energy increased the unique contribution of visual features and decreased that of action features, indicating that the action features share variance with motion energy. However, the three stages of temporal processing were preserved in the fixed-effects analysis even with the addition of motion energy, suggesting that the three feature groups made distinct contributions to the neural patterns. Importantly, the unique contribution of social-affective features was unchanged in both analyses by the addition of the motion energy model.”